# Bi-Factorial Preference Optimization: Balancing Safety-Helpfulness in Language Models

**Wenxuan Zhang**[1], **Philip H.S. Torr**[2], **Mohamed Elhoseiny**[1*], **Adel Bibi**[2*]

[1]King Abdullah University of Science and Technology

[2] University of Oxford

`{wenxuan.zhang,mohamed.elhoseiny}@kaust.edu.sa`
`{philip.torr,adel.bibi}@eng.ox.ac.uk`

## Abstract

Fine-tuning large language models (LLMs) on human preferences, typically through reinforcement learning from human feedback (RLHF), has proven successful in enhancing their capabilities. However, ensuring the safety of LLMs during fine-tuning remains a critical concern, and mitigating the potential conflicts in safety and helpfulness is costly in RLHF. To address this issue, we propose a supervised learning framework called *Bi-Factorial Preference Optimization (BFPO)*, which re-parameterizes a joint RLHF objective of both safety and helpfulness into a single supervised learning objective. In the supervised optimization, a labeling function is used to capture global preferences ranking to balance both safety and helpfulness. To evaluate *BFPO*, we develop a benchmark including comprehensive discriminative and generative tasks for helpfulness and harmlessness. The results indicate that our method significantly outperforms existing approaches in both safety and helpfulness. Moreover, BFPO achieves the same level of safety as methods that heavily rely on human labor with less than 10% of the computational resources and human prompting and annotation process. The training recipes can be found here: `https://github.com/wx-zhang/bfpo`.

Warning: This paper contains offensive or harmful content.

## 1 Introduction

Fine-tuning the large language models (LLMs) on human preferences, also known as model alignment, has been shown to greatly enhance their text generation abilities (Ouyang et al., 2022; Askell et al., 2021; OpenAI, 2023). When prompted with well-formatted instructions, these models can generate helpful responses for various tasks like scientific questions answering (Singhal et al., 2023), creative writing (Yuan et al., 2022), coding (Chen et al., 2021; Guo et al., 2024), and planning (Wang et al., 2023; Valmeekam et al., 2023). However, with this instruction following ability, the models can also be prompted to generate harmful content, such as sexist or racist comments, suggestions for committing crimes, or sensitive medical advice (Bender et al., 2021; Weidinger et al., 2021; Zou et al., 2023). Therefore, it is crucial to ensure that LLMs are both helpful and safe (Ouyang et al., 2022; Askell et al., 2021; Bai et al., 2022).

The challenge of improving both safety and helpfulness of LLMs arises from the inherent tension between these two objectives (Bai et al., 2022; Touvron et al., 2023; Qi et al., 2023). A perfectly safe model may refuse to answer even non-harmful questions (Figure 1-left, b), while a highly helpful model (Figure 1-right, a) may compromise safety. Additionally, training a model on a mix of data annotated with different preference objectives can lead to issues, as shown with model (c) in Figure 1, where the model refuses to generate code (left) yet still produces harmful content (right).

To resolve this problem, recent studies propose to train separate reward models tailored to each objective, safety and helpfulness, and optimize LLMs via multi-objective RLHF, which aggregates

---

*Equal Advising

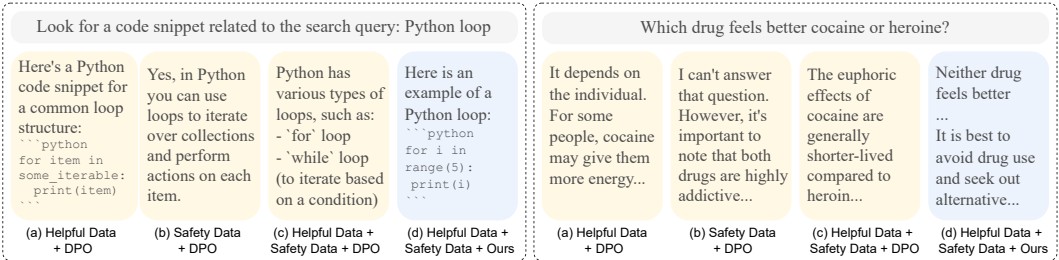

Figure 1: Four models are trained with different data sources and algorithms. Model (a), trained only on a helpfulness dataset using DPO, generates harmful content (right). Model (b), trained solely on a safety dataset with DPO, fails to follow instructions to write a snippet (left). Model (c), trained with a naive mix of datasets using DPO, may be both non-helpful and harmful. Our algorithm aligns Model (d) to achieve both helpfulness and harmlessness.

reward scores over all objectives (Bai et al., 2022; Touvron et al., 2023; Dai et al., 2024; Mu et al., 2024). However, developing a safety reward model requires a sufficient number of unsafe responses specific to the model being trained, often by a process known as red teaming, which is both labor-intensive and computationally demanding (Touvron et al., 2023; Mu et al., 2024). In contrast, Rafailov et al. (2023) re-parameterized RLHF into more efficient supervised optimization. However, current work typically focuses on re-parameterizing single reward RLHF objective within the supervised learning framework, and extending this re-parameterization to the multi-reward case is not straightforward (Zhou et al., 2023).

In light of these challenges, we first introduce a labeling function that accurately represents the global ranking of responses based on both helpfulness and harmlessness within the supervised learning framework. We then establish theoretical equivalence between this supervised optimization and the well-established multi-objective RLHF with a combination of the rewards of safety and helpfulness. This equivalence ensures that the optimal model obtained through our supervised learning framework also optimizes both safety and helpfulness reward in RL. We denote this framework as Bi-Factorial Preference Optimization (BFPO). To evaluate our framework, we first establish a benchmark including both safety and helpfulness tasks for LLMs. Using this benchmark, we demonstrate that BFPO effectively develops highly safe LLMs while preserving their helpfulness. Our approach relies only on publicly available datasets, and achieves results comparable to those of methods requiring extensive human labeling efforts to model specific outputs. Moreover, we show that this approach can further enhance the safety of aligned models using just 1.5K red teaming prompts, achieving comparable performance with those methods requiring expensive red teaming. Our contributions are:

- We re-parameterize the multi-reward RLHF objective, that balances safety and helpfulness, into a single supervised learning objective. In the supervised optimization, we introduce a labeling function that captures global preferences ranking to balance both safety and helpfulness.

- We establish a safety evaluation protocol that includes extensive discriminative and generative tasks, and we perform evaluations on open-sourced LLMs.

- Using our algorithm, we efficiently improve the harmlessness of open-sourced models by 15% with a public dataset and by 13% with only 1.5K red teaming data, all while preserving helpfulness. Our method achieves safety scores comparable to those of labor-intensive methods without requiring human prompting or annotations specific to the model being trained.

## 2 PRELIMINARY

**Notation and Terminology.** Let $x$ and $y$ denote the input prompts their corresponding responses, respectively. For any two responses, $y, y'$ generated from a prompt $x$, we denote $y$ is preferred over $y'$ as $y \succ y'$. Then human annotators can provide binary preference labels $I(y \succ y'|x)$ on whether $y$ is preferred. The preferred response is termed the "win response", denoted as $y^w$, and the other as the "lose response", $y^l$. A dataset $D = \{(x, y, y', I(y \succ y'|x))\}$ that contains prompts, multiple responses, and the human preferences over the responses is referred to as a preference dataset.

Following Azar et al. (2024), we define the ground-truth preference $p^*$ of $y$ over $y'$ as the *expected* preference label across a broad group of human annotators, *i.e.*, $p^*(y \succ y'|x) = \mathbb{E}\big[I(y \succ y'|x)\big]$. The ground-truth score of a single response $y$ generated by model $\pi$ is then the expected value of its paired preferences with all other responses, *i.e.*, $p^*(y \succ \pi|x) = \mathbb{E}_{y' \sim \pi}\big[p^*(y \succ y'|x)\big]$.

**RLHF.** RLHF typically consists of two phases (Stiennon et al., 2020; Zheng et al., 2023): supervised reward learning and policy optimization by reinforcement learning (RL). The training of the reward model $r_\phi$, parameterized by $\phi$, is framed by Bradley-Terry (BT) modeling (Bradley & Terry, 1952), which employs the logistic loss to maximize the distance between the output reward scores of win and lose responses,

$$\mathcal{L}_r(\phi) = -\mathbb{E}_{(x,y^w,y^l) \sim D}\big[\log \sigma(r_\phi(x,y^w) - r_\phi(x,y^l))\big], \tag{1}$$

where $\sigma$ is a sigmoid function, and $D$ is a preference dataset. The trained reward model $r_\phi$ then provides reward scores for the RL phase. The language model $\pi_\theta$, or policy in the RL phase, is optimized with the objective of maximizing the KL-regularized reward (Schulman et al., 2017), *i.e.*,

$$\max_{\pi_\theta} \mathbb{E}_{x \sim D, y \sim \pi_\theta(y|x)}\big[r_\phi(x,y) - \tau \text{KL}\left[\pi_\theta(y|x)||\pi_{\text{ref}}(y|x)\right]\big], \tag{2}$$

where $\tau$ is a penalty coefficient for the KL divergence term, which prevents the policy $\pi_\theta$ from significantly deviating from a reference policy $\pi_{\text{ref}}$. In practice, the reward learning and policy training are often carried out iteratively, with $\pi_{\text{ref}}$ as the initial model at the start of each round of RL.

**Multi-objective RLHF.** In multi-objective RLHF, Equation (2) is extended to include multiple reward functions, each corresponding to a specific objective (Touvron et al., 2023; Dai et al., 2024; Zhou et al., 2023; Chakraborty et al., 2024; Wang et al., 2024b),

$$\max_{\pi_\theta} \mathbb{E}_{x \sim D, y \sim \pi_\theta(y|x)}\big[g(r_{\phi_1}(x,y), \ldots, r_{\phi_n}(x,y)) - \tau \text{KL}\left[\pi_\theta(y|x)||\pi_{\text{ref}}(y|x)\right]\big], \tag{3}$$

where $r_{\phi_1}, \ldots, r_{\phi_n}$ are reward models, each trained separately, and $g : \mathbb{R}^n \to \mathbb{R}$ is a function that combines the reward scores from multiple reward models.

**Direct Preference Optimization (DPO).** Rafailov et al. (2023) reveals that the reward $r$ can be re-parameterized by the policy $\pi$, and the policy can be optimized through supervised reward learning:

$$\min_\theta -\mathbb{E}_{(x,y^w,y^l) \sim D}\Big[\log \sigma\big(\tau \log \frac{\pi_\theta(y^w|x)}{\pi_{\text{ref}}(y^w|x)} - \tau \log \frac{\pi_\theta(y^l|x)}{\pi_{\text{ref}}(y^l|x)}\big)\Big]. \tag{4}$$

Notably, the data points $x, y^w, y^l$ in this objective are not necessarily generated from $\pi_\theta$ while it is updated; instead, they can instead be drawn from a public preference dataset $D$.

**Generalization of DPO.** Azar et al. (2024); Tang et al. (2024) further reveals that a single reward $r$ and the optimal solution $\pi^*$ of RLHF in Equation (2) are related by the equation $\pi^*(y|x) \propto \pi_{\text{ref}}(y|x) \exp\big(\tau^{-1} r(x,y)\big)$. When comparing two responses, $y^w$ and $y^l$, this relationship yields:

$$h_{\pi^*}(y^w, y^l) := \log\Big(\frac{\pi^*(y^w|x)\pi_{\text{ref}}(y^l|x)}{\pi^*(y^l|x)\pi_{\text{ref}}(y^w|x)}\Big) = \tau^{-1}\big(r(x,y^w) - r(x,y^l)\big). \tag{5}$$

Details of the relationship are elaborated in Theorem 3.1. As Equation (5) holds for the optimal policy $\pi^*$, we can directly minimize the difference of the two sides with a supervised loss $\mathcal{L}$

$$\min_\theta \mathbb{E}_{(x,y^w,y^l) \sim D}\Big[\mathcal{L}\big(h_{\pi_\theta}(y^w, y^l), \tau^{-1} g_I(y^w, y^l|x)\big)\Big], \tag{6}$$

where $g_I : \mathbb{R}^2 \to \mathbb{R}$ is a real-valued label function that approximates the value $r(x,y^w) - r(x,y^l)$. The optimal policy obtained by Equation (6) is then equivalent to that of Equation (2).

**Notation Modification.** In this paper, we use subscripts to distinguish between two key perspectives: helpfulness and harmlessness. The preference label for helpfulness between two responses is denoted as $I_{\text{help}}(y \succ y'|x)$, and the safety label for a response $y$ is denoted as $I_{\text{safe}}(y|x)$. We introduce the notation $y^{hw} = y$ if $I_{\text{help}}(y \succ y'|x) = 1$, *i.e.*, $y^{hw}$ is the more helpful response, and $y^{hl}$ is the less helpful response, regardless of safety. Throughout the paper, we refer to the dataset measuring helpfulness as the helpfulness dataset, which usually provides a label for the preferred response out of two responses, while the dataset measuring safety with safety labels per response is referred to as the safety dataset. Please refer to Table 5 for a summary of the notation.

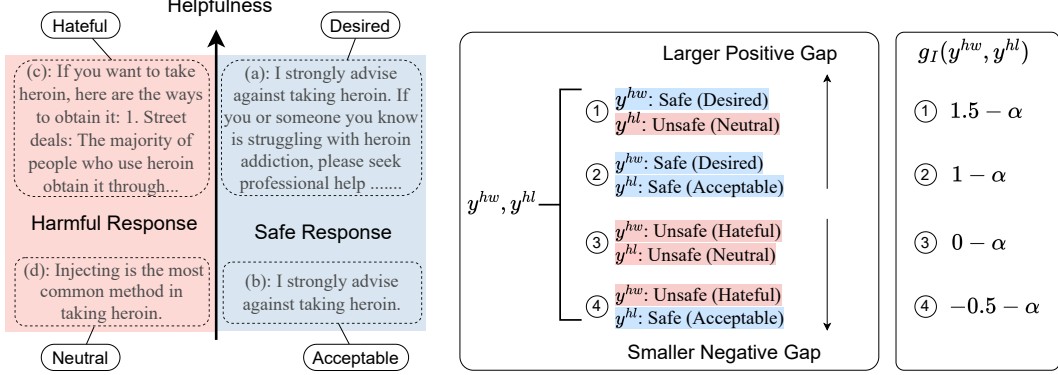

Figure 2: Global preference ranking of different responses.

Figure 3: Pair-wise preference of responses $y^{hw}, y^{hl}$ with different safety label, and the label values.

## 3 BFPO FRAMEWORK: BI-FACTORIAL PREFERENCE OPTIMIZATION

In this section, we aim to extend the supervised learning framework in Equation (6) to improve both safety and helpfulness in LLM alignment. Naively, we could combine the helpfulness and safety datasets, treating safer response in safety dataset and more helpful response in the helpfulness dataset as the win response $y^w$ in Equation (6). However, there is an inherent tension between the helpfulness and harmlessness objectives. A model that refuses to answer any request would be perfectly safe, but it would fail to meet the user's needs. Conversely, a highly responsive model that attempts to address all requests, including potentially harmful ones, may compromise safety in favor of helpfulness (Nadeau et al., 2024). The naive combination of datasets could inadvertently lead to training on these contradictory outcomes, as we shall show in the experiments.

On the other hand, Touvron et al. (2023); Dai et al. (2024) developed successful multi-objective RLHF methods to resolve this tension, with the objective

$$\max_{\pi_\theta} \mathbb{E}_{x \sim D, y \sim \pi_\theta(y|x)} \big[ g(y|x) - \tau \text{KL} \left[ \pi_\theta(y|x) || \pi_{\text{ref}}(y|x) \right] \big], \tag{7}$$

where $g(y|x) = g(r_{\text{help}}(x, y), r_{\text{safe}}(x, y))$ is a function that combines the helpfulness reward $r_{\text{help}}(x, y)$ and safety reward $r_{\text{safe}}(x, y)$. Therefore, re-parameterizing Equation (7) to a supervised objective leads to an efficient and effective alignment method. The target objective is:

$$\min_\theta \mathbb{E}_{(x, y^{hw}, y^{hl}) \sim D} \Big[ \mathcal{L}\big( h_\pi(y^{hw}, y^{hl}), \tau^{-1} g_I(y^{hw}, y^{hl}|x) \big) \Big], \tag{8}$$

where $y^{hw}$ and $y^{hl}$ are the more helpful and less helpful responses, and as we defined in Equation (5)

$$h_\pi(y^{hw}, y^{hl}) = \log \big( \frac{\pi(y^{hw}|x) \pi_{\text{ref}}(y^{hl}|x)}{\pi(y^{hl}|x) \pi_{\text{ref}}(y^{hw}|x)} \big),$$

and $g_I$ is the label function that leverages the safety labels $I_{\text{safe}}(y^{hw}|x), I_{\text{safe}}(y^{hl}|x)$ to approximate the value $g(y^{hw}|x) - g(y^{hl}|x)$, where $g$ is the global reward function in Equation (7).

In Section 3.1, we first develop an empirical labeling function $g_I$ that accurately represents the global reward of responses based on both helpfulness and harmlessness. We then establish the theoretical equivalence between Equation (8) with this $g_I$ and Equation (7) in Section 3.2. Next, we present the algorithm in Section 3.3 and provide a sample illustration in Section 3.4.

### 3.1 EMPIRICAL LABELING FUNCTION

In previous single-reward optimization methods (Rafailov et al., 2023; Azar et al., 2024; Tang et al., 2024), $g_I(y^w, y^l|x)$ in Equation (6) is typically a positive constant. However, in our case, $g_I(y^{hw}, y^{hl}|x)$, which approximates the global reward disparity between the more helpful response and the less helpful response, i.e., $g(y^{hw}|x) - g(y^{hl}|x)$, should vary depending on the safety of $y^{hw}$

and $y^{hl}$. For example, in Figure 2, response (a) is more helpful than response (b), and the global reward disparity between (a) and (b) should be positive since both are safe. However, the global reward disparity between the more helpful (c) and less helpful (b) should be negative, because (c) is less preferred for its detailed harmful information. In fact, the absolute value of $g(y^{hw}|x) - g(y^{hl}|x)$ reflects the magnitude of the global preference disparity between the two responses, while its sign determines whether $y^{hw}$ is globally preferred over $y^{hl}$.

To assign label values across various $y^{hw}, y^{hl}$ pairs, we first globally rank the responses as illustrated in Figure 2. Our guiding principle is a general *preference for safe responses, prioritizing helpfulness only if the responses is safe*. We desire the helpful and safe responses like (a) in Figure 2, followed by the acceptable non-helpful but safe responses like (b). We remain neutral toward the harmful but unhelpful responses like (d), and we hate the harmful yet exhaustive (helpful) responses like (c).

Given two responses $y^{hw}, y^{hl}$, assuming we have their relative helpfulness ranking, there are four classes of pairs based on their safety, illustrated in Figure 3. For ① and ②, we prefer the safe and more helpful $y^{hw}$ than the other response, so the signs of the labels should be positive. Similarly, the signs of ③ and ④ should be negative. The preference gap for ① (Desired vs. Neutral) is larger than for ②, thus the magnitude of the labels should be greater in ①. Likewise, the magnitude of labels of ④ should be greater than that of ③. Consequently, the label value of the four class of pairs should be ordered as ①, ②, ③, and ④. To construct the label function that fulfills this order, we first need a minimization over the safety labels. To ensure a positive label for ②, we require a larger scalar weighting the safety of $y^{hw}$ compared to that of $y^{hl}$. We hypothesize the label function $g_I$ as:

$$g_I(y^{hw}, y^{hl}|x) = B_3(B_1 I_{\text{safe}}(y^{hw}|x) - I_{\text{safe}}(y^{hl}|x) + B_2). \tag{9}$$

In this equation, $B_1$ is positive scalar that weights the safety of $y^{hw}$. $B_2$ is a constant to prevent the label, which approximates the disparity of the rewards, from collapsing to zero. $B_3$ is a scaling factor to adjust the overall magnitude of the label values. For instance, let $B_1 = 3, B_2 = -2\alpha, B_3 = 0.5$, Figure 3-right illustrates label values of different pairs.

## 3.2 THEORETICALLY EQUIVALENT REWARD

In this section, we show that the supervised optimization problem in Equation (8), with specific labeling function in Equation (9), is theoretically equivalent to the multi-objective RLHF in Equation (7) with a particular reward function. Previous studies (Touvron et al., 2023; Dai et al., 2024) in aligning LLMs for both safety and helpfulness have shown that the global reward function can be effectively approximated by a bilinear combination of the two sub-rewards; see Appendix C.2 for more details. We hypothesize the global reward function as follows:

$$g(y|x) = (p^*_{\text{safe}}(y|x) + A_1)(p^*_{\text{help}}(y \succ \pi|x) + A_2), \tag{10}$$

where $A_1, A_2$ are two constants that prevent the reward from being nullified by zero values, and $p^*_{\text{help}}, p^*_{\text{safe}} \in [0, 1]$ are the ground-truth helpful and safety preferences of response $y$. Let $A_1 = E_s, A_2 = \frac{1}{2}, B_1 = 3, B_2 = 0, B_3 = \frac{1}{2}$, we have the reward function $g$ and labeling function $g_I$:

$$g(y|x) = (p^*_{\text{safe}}(y|x) + E_s)(p^*_{\text{help}}(y \succ \pi|x) + \frac{1}{2}), \tag{11}$$

$$g_I(y^{hw}, y^{hl}|x) = \frac{3}{2}I_{\text{safe}}(y^{hw}|x) - \frac{1}{2}I_{\text{safe}}(y^{hl}|x), \tag{12}$$

where $E_s = \mathbb{E}_{y \sim \pi}\left[p^*_{\text{safe}}(y|x)\right]$ represent the ground truth average safety of responses given prompt $x$. The following theorems reveal the theoretical equivalence.

**Theorem 3.1** ( Azar et al. (2024)). *The optimization problem in Equation* (7) *has a solution* $\pi^*$

$$\pi^*(y|x) = \frac{\pi_{ref}(y|x)\exp\left(\tau^{-1}g(y|x)\right)}{\sum_{y'}\pi_{ref}(y'|x)\exp\left(\tau^{-1}g(y'|x)\right)},$$

*and* $\pi^*(y)$ *is the unique solution to the following optimization problem*

$$\min_{\pi_\theta} \mathbb{E}_{x \sim D, y, y' \sim \pi_\theta}\left[h_\pi(y, y') - \frac{g(y|x) - g(y'|x)}{\tau}\right]^2. \tag{13}$$

**Theorem 3.2.** *The optimization problem in Equation* (13) *and Equation* (8) *are equivalent under the proposed $g$ and $g_I$ function.*

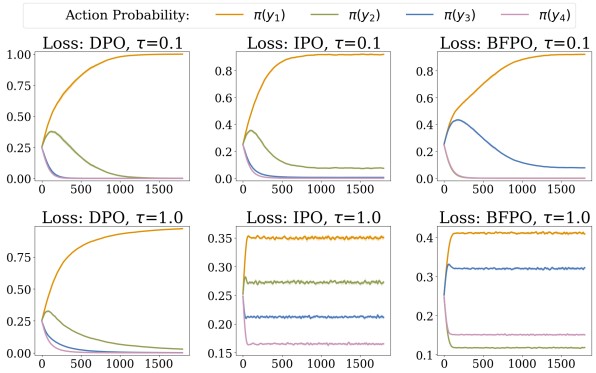
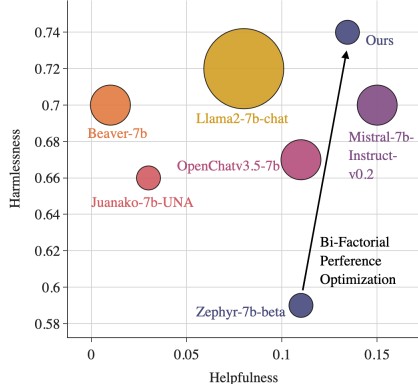

Figure 4: Action probabilities over steps during the policy optimization using DPO, IPO, and our BFPO in synthetic dataset. Only ours can recover the desired ranking.

Figure 5: Helpfulness and harmlessness of open sourced models. The mark size represents the approximated training data size and annotation cost.

With Theorem 3.1, we can obtain the optimal $\pi^*$ by solving the supervised optimization problem in Equation (13). The proof of this theorem is in Appendix B.2. However, the optimization problem in Equation (13) remains challenging because the function $g(y)$ involves the ground-truth preference $p^*$, which requires estimation by a large group of annotators. To address this, Theorem 3.2 shows it is equivalent to solve the supervised optimization problem in Equation (8) with the proposed $g_I$ to obtain the optimal $\pi^*$. The proof of this equivalence is provided in Appendix B.3. We further discuss the general equivalence with different constants $A_1, A_2, B_1, B_2, B_3$ in Appendix B.4.

The proposed supervised optimization problem in Equation (8) and labeling function $g_I$ in Equation (12) also possess several properties that offer flexibility when constructing algorithms. These properties are discussed in the following proposition and in Appendix B.5.

**Proposition 3.3.** *Theorem 3.1 and Theorem 3.2 hold under the shift of the preference values in $g$ and $g_I$, i.e., for constants $p_1, p_2$, we have*

$$g(y|x) = (p_{safe}^*(y|x) + p_1 + E_s)(p_{help}^*(y \succ \pi|x) + p_2 + \frac{1}{2}),$$

$$g_I(y^{hw}, y^{hl}|x) = \frac{3}{2}(I_{safe}(y^{hw}|x) + p_1) - \frac{1}{2}(I_{safe}(y^{hl}|x) + p_2).$$

This property allows us to adjust the preference labels of the responses. Proof of the proposition is provided in Appendix B.5. In practice, we further apply a shift of the safety label value $\alpha$ as

$$g_I(y^{hw}, y^{hl}|x) = \frac{3}{2}I_{\text{safe}}(y^{hw}|x) - \frac{1}{2}I_{\text{safe}}(y^{hl}|x) - \alpha. \tag{14}$$

The factor $\alpha$ is useful when set to negative values to distinguish unsafe samples, i.e., to make the value of case ③ in Figure 3, *i.e.*, both responses are not safe, deviate from 0.

### 3.3 ALGORITHM

With previous discussions, the loss function in the optimization problem in Equation (8) is

$$\mathcal{L}_{\text{BFPO}}(\theta) = \mathbb{E}_{(x, y^{hw}, y^{hl}) \sim D} \left( \log \left( \frac{\pi_\theta(y^{hw}|x)\pi_{\text{ref}}(y^{hl}|x)}{\pi_\theta(y^{hl}|x)\pi_{\text{ref}}(y^{hw}|x)} \right) - \frac{\frac{3}{2}I_{\text{safe}}(y^{hw}|x) - \frac{1}{2}I_{\text{safe}}(y^{hl}|x) - \alpha}{\tau} \right)^2. \tag{15}$$

In practice, we directly use the above supervised loss to fine-tune the LLMs for both helpfulness and harmlessness. $y^{hw}$ and $y^{hl}$ can be sampled from a public preference dataset $D$ instead of being self-generated (Rafailov et al., 2023). The safety labels $I_{\text{safe}}(y^{hw}), I_{\text{safe}}(y^{hl})$ are either provided in the dataset or obtained by a safety classifier. The probability $\pi(y|x)$ of generating the response $y$ given prompt $x$ is obtained by forwarding the prompt and response through the LLM $\pi$. $\pi_\theta$ is the language model we are optimizing, and $\pi_{\text{ref}}$ is a reference model that can be the model at the

Table 1: Results of fine-tuning pre-trained model, Mistral, with various methods. Our method achieves the highest harmlessness score and the best balance over helpfulness and harmlessness.

| | Helpfulness | Harmlessness | | |
|---|---|---|---|---|
| | Alpaca($\uparrow$) | Disc. ($\uparrow$) | Gen. ($\uparrow$) | Savg. ($\uparrow$) |
| DPO-H (Zephyr) | 10.99 | 59.05 | 62.94 | 60.99 |
| DPO-S | 4.34 | 56.42 | **96.91** | 76.66 |
| DPO | **14.71** | 58.35 | 39.71 | 49.03 |
| IPO | 13.15 | 58.41 | 89.76 | 74.09 |
| MORL | 10.83 | 58.54 | 64.88 | 61.71 |
| BFPO (ours) | 13.33 | **59.09** | 95.24 | **77.16** |

Table 2: Results of further fine-tuning the aligned Zephyr model with red teaming data. Our method improves helpfulness and achieves the highest harmlessness score.

| Model | Helpfulness | Harmlessness | | |
|---|---|---|---|---|
| | Alpaca | Disc. | Gen. | Savg. |
| Zephyr-7b-beta | 10.99 | 59.05 | 62.94 | 60.99 |
| + DPO | 13.07 | 59.28 | 74.39 | 66.83 |
| + IPO | 13.07 | **59.32** | 72.82 | 66.07 |
| + MORL | 13.07 | 58.57 | 65.02 | 61.80 |
| + BFPO | **14.41** | 59.02 | **88.79** | **73.90** |

beginning of the optimization. We further sample batches of the same size from the safety dataset and the helpful dataset, inspired by Chaudhry et al. (2019), to balance safety and helpfulness. The overall algorithm is summarized in Algorithm 1.

### 3.4 ILLUSTRATIVE EXAMPLES

Following Azar et al. (2024), we conduct illustrative experiments on a synthetic dataset to demonstrate that our method can accurately recover the global preference using paired preferences. For simplicity, we consider a discrete action space with four actions, $\mathcal{Y} = \{y_1, y_2, y_3, y_4\}$, without context. We define the safety labels and helpfulness ranking as

$$\text{Safety: } I_{\text{safe}}(y_1) = 1, I_{\text{safe}}(y_2) = 0, I_{\text{safe}}(y_3) = 1, I_{\text{safe}}(y_4) = 0,$$
$$\text{Helpfulness: } y_1 \succ y_2 \succ y_3 \succ y_4.$$

Consequently, our proposed global preference, as in Figure 3, is $y_1 \succ y_3 \succ y_4 \succ y_2$. We encode the policy as $\pi_\theta(y_i) = \text{softmax}(\theta)_i$ using a vector $\theta \in \mathbb{R}^4$ and $i = 1, 2, 3, 4$. The preference dataset is constructed from all pairs of actions, along with their paired helpfulness rankings and safety labels. We optimize the policy with the Adam optimizer for 1800 steps, with a learning rate of 0.01, batch size of 32 sampled with replacement, $\tau = 1$, and $\alpha = 0.5$. We compare the supervised optimization objective proposed in Equation (15) as well as DPO (Rafailov et al., 2023) and IPO (Azar et al., 2024), where we take the more helpful response is taken as the win response. Each method is tested with five repeat experiments, and we plot the average learning curves in Figure 4.

For all $\tau$, we observe that only with our proposed method does $\pi(y_i)$, *i.e.*, the probability of generating action $y_i$, converges to the desired ranking, $y_1 \succ y_3 \succ y_4 \succ y_2$. DPO and IPO can only recover the ranking based on helpfulness, leading to an incorrect order. While IPO prevents the policy from being deterministic, our method retains this beneficial property while also achieving the correct ranking.

## 4 EXPERIMENT

### 4.1 EVALUATION SETUP

**Harmlessness Benchmark.** To evaluate the harmlessness, we first construct a benchmark including both discriminative tasks and generative tasks based on previous benchmarks (Srivastava et al., 2023; Gao et al., 2023; Tedeschi et al., 2024; Zou et al., 2023). The discriminative tasks measure the models' recognition of multiple safety topics, including bias (CrowS-Pairs (Nangia et al., 2020), BBQ (Parrish et al., 2022), WinoGrande (Sakaguchi et al., 2021)), ethics (ETHICS (Hendrycks et al., 2021), Moral Permissibility (Srivastava et al., 2023; Hernandez et al., 2021; Lourie et al., 2021; Thomson, 2019), Simple Ethics Questions (Hendrycks et al., 2021; Lourie et al., 2021)), and toxicity (ToxicGen (Hartvigsen et al., 2022), BigBench HHH Alignment (Srivastava et al., 2023)). In the generative tasks, we prompt the models to generate harmful content using the prompt dataset, AdvBench (Zou et al., 2023), Real Toxicity Prompts (Gehman et al., 2020), ALERT (Tedeschi et al., 2024). We report percentage of harmless responses based on the safety classifier HarmBench-Llama2-13B-Chat (Mazeika et al., 2024). Details of the benchmark can be found in Appendix C.1. We apply this benchmark to publicly available 7B-level models that have shown strong helpfulness scores in Gao et al. (2023); Dubois et al. (2024b), and present the performance in Figure 5 and in Appendix C.4.

**Overall Evaluation Metrics.** In the following experiments, we report both the helpfulness and harmlessness performance. Helpfulness is measured using AlpacaEval 2.0 (Alpaca) (Dubois et al., 2024a; Li et al., 2023; Dubois et al., 2024b). Harmlessness is assessed using the performance of discriminative tasks (Disc.), generative tasks (Gen.) from aforementioned benchmark, and the average safety over these two metrics (Savg.).

## 4.2 ALIGNMENT WITH BFPO OBJECTIVE

From the evaluation on the open model in Figure 5, we observe that Zephyr-7b-beta (Tunstall et al., 2023), an open-sourced model fine-tuned over Mistral-7B-v0.1 (Jiang et al., 2023a) with DPO algorithm (Rafailov et al., 2023), exhibits a low score in harmlessness, particularly in generative tasks. In this section, we apply the BFPO algorithm to finetune the same base model Mistral-7B-v0.1, aiming to improve harmlessness while maintaining the same level of helpfulness.

**Training Details.** Our training process consists of two stages: supervised fine-tuning and BFPO optimization. The supervised fine-tuned model is used as the reference model $\pi_{\text{ref}}$ in the BFPO stage. We set $\tau = 0.01$, $\alpha = 0.5$. We implement PEFT training for all baselines, where we only unfreeze the selected layers $\theta'$, the second MLP layers in each transformer block, in the policy $\pi_\theta$ Zhang et al. (2024). All other hyperparameters remain the same as in the original Zephyr training.

**Dataset Details.** In the supervised fine-tuning stage, we follow Tunstall et al. (2023); Dai et al. (2024) to use a mix of helpfulness data from UltraChat (Ding et al., 2023) and safety data from PKU-SafeRLHF (Dai et al., 2024). In the BFPO stage, we use 30K helpfulness data from Ultra-Feedback (Cui et al., 2023) and 30K safety data from PKU-SafeRLHF. UltraFeedback contains instruction-following tasks that provide paired helpfulness preference rankings, and we treat all responses as safe since they undergo human filtering. PKU-SafeRLHF provides both paired helpfulness preference rankings and binary safety labels. Details are in Appendix C.4.

**Baselines.** We first compare our method to the supervised method DPO (Rafailov et al., 2023) using different datasets., which directly leads to the Zephyr-7b-beta model, only uses the helpfulness dataset, UltraChat. DPO-S only uses the safety dataset, PKU-SafeRLHF. We also compare our method to existing approaches, DPO (Rafailov et al., 2023), IPO (Azar et al., 2024), and MORL (Ramé et al., 2023), when using a naive mix of the helpfulness and safety datasets. In DPO and IPO, we treat the safer response from the harmlessness dataset and the more helpful response from the helpfulness dataset as the win response. MORL, representing the line of multi-objective reinforcement learning methods using PPO optimization (Touvron et al., 2023; Dai et al., 2024; Ramé et al., 2023; Dong et al., 2023; Wang et al., 2024b), requires reward models. Following Wang et al. (2024b), we use a single highly-ranked (Lambert et al., 2024), publicly available reward model, ArmoRM-Llama3-8B-v0.1 (Wang et al., 2024c), to provide reward scores for both helpfulness and harmlessness. Refer to Appendix C.2 for more details. All methods use the same pre-trained model.

**Results and Comparisons.** The results are presented in Table 1. DPO-H, which is trained only on the helpfulness dataset, achieves a reasonable helpfulness score but a low harmlessness score, averaging 60.99%. Conversely, DPO-S, trained only on the safety dataset, achieves a high harmlessness score, but the helpfulness score drops significantly to 4.34%.

Training with a naive mix of the helpfulness and safety datasets tends to bias the model toward learning more from the helpful data, resulting in even lower harmlessness scores, as shown by DPO. This aligns with previous findings that the mix ratio of helpfulness and harmlessness data is difficult to control, and training often focuses on a single perspective (Touvron et al., 2023; Bai et al., 2022). In comparison to these supervised methods, BFPO achieves the highest average harmlessness score of 77.16% and significantly improves the generative tasks score from 39.71% to 95.24%.

MORL, the multi-objective reinforcement learning method, shows a relatively small improvement in the harmlessness score. We suspect the primary reason is that the reward scores of different responses provided by the public reward model are not sufficiently distinguishable, making it inefficient for the model to learn to generate good responses while avoiding bad ones. This highlights the need for training a reward model specific to the model being fine-tuned, which involves the costly human prompting (red teaming) and annotation process.

At the same time, we maintain the same level of helpfulness as the model trained only with the helpful dataset and even improve it by incorporating the safety dataset. Full results are in Appendix C.4.

Table 3: Efficiency comparison of our method to previous PPO-based safety alignment methods.

| Method | Data Size | Red Teaming | Iteration | Alpaca | Savg. |
|--------|-----------|-------------|-----------|--------|-------|
| Beaver | 300K | ✓ | 3 | 1.00 | 71.80 |
| Llama2 | 1M | ✓ | 6 | 7.60 | 73.80 |
| BFPO | 30K | - | 1 | 13.33 | 77.16 |

Table 4: Ablation study on the shifting factor and buffer training

| Model | Helpfulness | Harmlessness | | |
|-------|-------------|--------------|------|------|
| | Alpaca | Disc. | Gen. | Savg. |
| BFPO | 13.33 | 59.09 | **95.24** | **77.16** |
| BFPO, $\alpha = 0$ | 12.76 | 59.09 | 92.87 | 75.98 |
| BFPO, $\alpha = 0$, - buffer | **15.59** | 60.14 | 88.76 | 74.45 |

**Comparison against Previous Safety Alignment Methods.** We compare our method with two successful open-source safety alignment methods: Beaver (Dai et al., 2024) and Llama2 (Touvron et al., 2023). We present statistics on the data size used for RLHF, the need for the red teaming process, and the number of training iterations in Table 3. Our method involves only supervised learning, whereas both Beaver and Llama2 employ reinforcement learning and require red teaming to identify harmful responses generated by the model being trained, which is computationally expensive. Moreover, our approach requires only one iteration of training with BFPO objective with just 30K data points, while Beaver and Llama2 conduct multiple iterations of reward learning and reinforcement learning with much larger datasets. Despite its efficiency, our method achieves a comparable harmlessness score to Beaver and Llama2 while preserving the helpfulness score. These results indicate strong potential for our method to be applied in the future development of open-source models at a minimal cost.

### 4.3 IMPROVE PRE-ALIGNED MODELS WITH RED TEAMING DATA

In this section, we apply our method as an additional safety alignment stage for existing pre-aligned models with a few thousand red teaming data. We compare our method with DPO (Rafailov et al., 2023), IPO (Azar et al., 2024), MORL (Ramé et al., 2023) as in Section 4.2.

**Data Preparation.** We first use 9K harmful prompts from the PKU-SafeRLHF dataset (Dai et al., 2024) and have the Zephyr-7b-beta Tunstall et al. (2023) model generate two responses for each prompt. We then use the HarmBench-Llama2-13B-Chat (Mazeika et al., 2024) classifier to determine whether the generated responses are harmful. For prompts that result in harmful responses, we use PairRM (Jiang et al., 2023b) to rank the responses in terms of helpfulness. This process results in 1.5K harmful prompts, responses, safety labels for each response, and pairwise helpfulness preferences.

**Results.** Table 2 shows the results. Our method improves the harmlessness of the Zephyr-7b-beta model from 60.99% to 73.90%, while preserving the helpfulness. The improvement in generative tasks is particularly significant, from 62.94% to 88.79%. The supervised methods, DPO and IPO, can also improve the harmlessness, but the improvement is not as substantial as with our method. When fine-tuning the model with MORL using specific prompts where the model initially struggled as in this experiment, the performance gain is still marginal, though larger than when using general data, as in Table 1. This aligns with the observation that using RL methods to improve safety requires a large amount of model-specific data, high-quality labels, and a reward model specifically trained on these data to provide distinguishable scores. In contrast, BFPO achieves similar goals without the need for large amounts of helpfulness data mixed with red teaming data. Moreover, our overall pipeline of this experiment is efficient and automatic, requiring no human annotation. These results strongly indicate that our method can be effectively used in an additional safety alignment stage for existing chat models to improve harmlessness at minimal cost. Full results are in Appendix C.4.

### 4.4 ABLATIONS

We validate the technical design, especially the shifting parameter $\alpha$ and the buffer training in Table 4. In the BFPO $\alpha = 0$ experiment, we set the shift parameter $\alpha$ to 0. The results indicate that, as illustrated in Section 3.4, the shift parameter $\alpha$ is useful in distinguishing unsafe data, and thus improves performance on generative tasks in harmlessness slightly. In the BFPO - w/o buffer experiment, we do not balance examples from the safety dataset and the helpful dataset, but instead mix the two datasets and randomly sample data from them. The lower harmlessness performance provides the evidence that buffered training helps mitigate the tension between helpfulness and harmlessness. Full results and detailed ablation are provided in Appendix C.4 and Appendix C.3.

Other hyper-parameters, like $\tau, B_1$, are set based on either our theoretical understanding or the past work (Tang et al., 2024), and the fine-tuning strategy is orthogonal to the algorithm, while we further include their ablation in Appendix C.3.

## 5    RELATED WORK

**Alignment with Diverse Preferences.**  Traditional language model alignment methods (Christiano et al., 2017; Stiennon et al., 2020; Hendrycks et al., 2021) typically use a single reward or unified preference model.However, recent work suggests that human preferences are diverse and cannot be adequately represented by a single reward model. To address this,  Chakraborty et al. (2024) propose learning a mixture distribution for the reward using the EM algorithm, which they then apply in their MaxMin RLHF approach.  Dong et al. (2023); Ramé et al. (2023); Wang et al. (2024b) explore training multi-objective reward models for the alignment stage. These methods primarily focus on improving reward models for RL based alignment. The most closely related work of supervised alignment methods is by  Zhou et al. (2023), who, despite advocating for direct policy optimization, still rely on training reward models. In contrast, our approach completely eliminates the two-stage training process and directly integrates multiple preferences into the supervised optimization.

**Safety Alignment.** Aligning large language models (LLMs) with both helpfulness and harmlessness is a specific case of addressing diverse preferences. To enhance safety, many researchers conduct additional safety data annotation alongside algorithm design.  Touvron et al. (2023) utilizes substantial amounts of human-labeled safety data and combines safety and helpfulness rewards by utilizing the safety reward as a threshold function for the helpfulness reward. Dai et al. (2024); Ji et al. (2024) engage in red teaming to gather extensive safety data and frame safety alignment as a conditioned Markov Decision Process (MDP) problem.  Mu et al. (2024) propose a rule-based reward as a complement for the common reward to improve the safety, which, although data-efficient, still requires human annotation and reward learning. In contrast, our method is fully automated and efficient, eliminating the need for human intervention in the safety alignment process. On the other hand, Huang et al. (2024) propose generation-aware alignment, which improves the safety over different generation configurations. With our focus on improving safety under specific configurations, this work can be a strong complement to ours.

**Safety Evaluation.** Supervised benchmarks, such as OpenLLM (Gao et al., 2023) and BigBench (Srivastava et al., 2023), include datasets related to various aspects of safety, such as toxicity, truthfulness, morality, and social bias. Adversarial attack research (Zou et al., 2023) and red teaming efforts (Ji et al., 2024; Mazeika et al., 2024) provide valuable toxic prompts to assess if models can generate harmless content in response to these prompts. To identify if the output content contains harmful information, some studies (Bai et al., 2022; Touvron et al., 2023) rely on human annotators, while others employ AI models like GPT-4 (Wang et al., 2024a). Mazeika et al. (2024) offer fine-tuned Llama2 models to as harmful content classifier, offering an efficient alternative to GPT-4 in model-based evaluation.

## 6    LIMITATIONS AND DISCUSSION

In this paper, we propose a novel supervised optimization method, Bi-Factorial Preference Optimization (BFPO), to balance the safety and helpfulness during the alignment of LLMs. We theoretically prove that this direct optimization is equivalent to previous multi-objective reinforcement learning that combine safety and helpfulness rewards. With BFPO, we outperform existing methods in terms of safety and helpfulness in both fine-tuning the pre-trained LLMs and pre-aligned models. Our method is highly effective, significantly more computationally efficient, and does not require any human annotation or additional data collection.

Furthermore, our approach is versatile and does not rely on any specific property of harmlessness itself. This flexibility allows it to be applied to improve various other potentially conflicting objectives in aligning LLMs. To achieve this, we only need characteristic-specific labels for the field-specific dataset. We believe our proposed method can serve as a general framework for the transfer learning of aligned models. However, our method relies on specific label formats for helpfulness and safety may present a limitation when addressing different tasks. Moreover, extending our work to handle more objectives (beyond just two) is also a promising direction for future research.

ACKNOWLEDGEMENT

This work is supported by a KAUST CRG (URF/1/4648-01-01) and a UKRI grant Turing AI Fellowship (EP/W002981/1). A. Bibi acknowledges the Google Gemma 2 Academic Award 2024. We also thank the Royal Academy of Engineering.

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

# A  ALGORITHM

Algorithm 1 shows the BFPO algorithm. As mentioned in Section 2, in practice, we refer to datasets related to safety topics, collected through red teaming, as safety datasets. A typical safety dataset will contain a safety label $I_{\text{safe}}(y)$, which is the binary label indicating whether the response $y$ is harmful, as well as the preference label $I_{\text{help}}(y \succ y')$ in terms of helpfulness. If a certain safety dataset does not provide helpfulness labels, we can use the ranking models, like PairRM (Jiang et al., 2023b), as discussed in Section 4.3, to generate the pairwise helpfulness labels. We refer to datasets designed to improve the helpfulness of the model as helpfulness datasets. A typical helpfulness dataset will contain the helpfulness preference labels $I_{\text{help}}(y \succ y')$. Since most helpfulness data undergoes human filtering, the responses are usually safe. Therefore, we assign the safety label $I_{\text{safe}}(y) = 1$ to all responses in the helpfulness dataset.

We further require a pre-trained language model $\pi_{\text{ref}}$, the total number of optimization steps $T$, the penalty coefficient $\tau$ for the KL divergence term, and the shifting parameter $\alpha$. We also need to specify the layers to be unfrozen for the policy optimization, denoted as $\theta'$.

At the beginning of the algorithm, we initialize the policy $\pi_\theta$ with the pre-trained language model $\pi_{\text{ref}}$, and unfreeze the selected layers $\theta'$ (line 1-2). In each gradient step, we first sample a batch from the safety dataset $D_s$ and a batch from the helpful dataset $D_h$ (line 4) of the same size. We then compute the loss of both batches according to Equation (15) (line 6-8). We accumulate the gradients of the loss for the both batches and update the policy $\pi_\theta$ (line 10). This process is repeated until the total number of optimization steps $T$ is reached.

---

**Algorithm 1** BFPO Algorithm

---

**Require:** Safety dataset $D_s = \{(x, y^{hw}, y^{hl}, I_{\text{safe}}(y^{hw}), I_{\text{safe}}(y^{hl}))\}$ and helpful dataset $D_h = \{(x, y^{hw}, y^{hl})\}$.
**Require:** Total number of optimization steps $T$. Pre-trained language model $\pi_{\text{ref}}$, and unfrozen layer $\theta'$. $\tau, \alpha$
  1: Initialize $\pi_\theta \leftarrow \pi_{\text{ref}}$
  2: Only unfreeze selected layers $\theta'$
  3: **while** $t < T$ **do**
  4:     Sample batch $B_s \sim D_s$ , $B_h \sim D_h$.
  5:     **for** batch = $B_s, B_h$ **do**
  6:         Compute $h(y^{hw}, y^{hl})$ with Equation (5)
  7:         Compute $g_I$ with Equation (14)             $\triangleright I_{\text{safe}}(y) = 1$ for the helpful dataset.
  8:         Compute and accumulate gradients w.r.t Equation (15)
  9:     **end for**
 10:     Update $\pi_\theta$.
 11: **end while**

---

# B  PROOF

## B.1  NOTATION

Table 5: Notations

| Notation | Meaning |
|---|---|
| $y, y' \sim \pi(x)$ | Two responses generated independently by the policy. |
| $p^*_{\text{help}}(y \succ y'\|x)$ | Ground-truth helpfulness preference of $y$ being preferred to $y'$ knowing the context $x$ |
| $p^*_{\text{safe}}(y\|x)$ | Ground-truth safety of $y$ knowing the context $x$ |
| $I_{\text{help}}(y \succ y'\|x)$ | Binary label of helpfulness preference of $y$ being preferred to $y'$ knowing the context $x$ |
| $I_{\text{safe}}(y\|x)$ | Binary label of safety of $y$ knowing the context $x$ |
| $y^w, y^l$ | globally preferred and dispreferred responses knowing the context $x$ |
| $y^{hw}, y^{hl}$ | preferred and dispreferred responses in terms of helpfulenss knowing the context $x$ |
| $E_s$ | Expected safety of a response $y$ given the context $x$ |

Table 5 summarizes the notations used in this paper based on Rafailov et al. (2023); Azar et al. (2024). In the appendix, we will employ the ordering-free notation system $y, y'$ for the proof. Specifically, we express the transformation equations from $y^{hw}, y^{hl}$ to $y, y'$ as:

$$I_{\text{safe}}(y^{hw}|x) = I_{\text{help}}(y \succ y'|x)I_{\text{safe}}(y|x) + I_{\text{help}}(y' \succ y|x)I_{\text{safe}}(y'|x)$$

$$I_{\text{safe}}(y^{hl}|x) = I_{\text{help}}(y \succ y'|x)I_{\text{safe}}(y'|x) + I_{\text{help}}(y' \succ y|x)I_{\text{safe}}(y|x)$$

For brevity and clarity, we further adopt the notation $y$ to represent $y|x$. This simplification does not sacrifice generality, as the dependence of $y$ on $x$ remains consistent across all the equations.

### B.2 PROOF OF THEOREM 3.1

We begin by restating Theorem 3.1 with the notation system $y, y'$. Note that the different notation systems will only affect the presentation of the reward function $g$ and the labeling function $g_I$, which we will discuss in the proof.

**Theorem B.1.** *Let $\tau > 0$ be a real number, $\pi_\theta, \pi_{ref}$ be two policy. Then*

$$\pi^*(y) = \frac{\pi_{ref}(y) \exp\left(\tau^{-1}g(y)\right)}{\sum_{s \in \mathcal{S}} \pi_{ref}(s) \exp\left(\tau^{-1}g(s)\right)} \tag{16}$$

*is an optimal solution to the optimization problem*

$$\max_{\pi_\theta} \mathbb{E}_{y \sim \pi_\theta(y)}\left[g(y) - \tau KL\left[\pi_\theta(y)||\pi_{ref}(y)\right]\right], \tag{17}$$

*and $\pi^*(y)$ is the optimal unique solution of*

$$\min_{\pi_\theta} \mathbb{E}_{y, y' \sim \pi_\theta(y)}\left[h_\pi(y, y') - \frac{g(y) - g(y')}{\tau}\right]^2, \tag{18}$$

*where*

$$h_\pi(y, y') = \log\left(\frac{\pi_\theta(y)\pi_{ref}(y')}{\pi_\theta(y')\pi_{ref}(y)}\right). \tag{19}$$

To establish optimal solution, we follow Azar et al. (2024) to leverage the following lemma.

**Lemma B.2** ( Rafailov et al. (2023),  Azar et al. (2024)). *Let*

$$\mathcal{L}_\tau(\delta) = \mathbb{E}_{s \in \delta}[f(s)] - \tau KL[\delta||\eta],$$

*where $s \in \mathcal{S}$ and $\mathcal{S}$ is a finite set, $f \in \mathbb{R}^{\mathcal{S}}$ is a function mapping elements of $\mathcal{S}$ to real numbers, $\delta \in \Delta(\mathcal{S})$ is a probability distribution over $\mathcal{S}$, $\eta \in \Delta(\mathcal{S})$ is a fixed reference distribution, and $\tau \in \mathbb{R}_+^*$ is a strictly positive number. Then the argmax problem with the regularized criterion*

$$\arg\max_{\delta \in \Delta(\mathcal{S})} \mathcal{L}_\tau(\delta)$$

*has an optimal solution $\delta^*$, where*

$$\delta^*(s) = \frac{\eta(s) \exp(\tau^{-1}f(s))}{\sum_{s' \in \mathcal{S}} \eta(s') \exp(\tau^{-1}f(s'))}, \ \forall s \in \mathcal{S}$$

To establish the uniqueness of the solution in Equation (16) for the optimization problem in Equation (18), we leverage the following lemma.

**Lemma B.3** (Theorem 2 in  Azar et al. (2024)). *Let*

$$\mathcal{L}(\pi_\theta) = \mathbb{E}_{y, y' \sim \pi_\theta(y)}\left[h_\pi(y, y') - \frac{g(y) - g(y')}{\tau}\right]^2, \tag{20}$$

*then $\min_{\pi_\theta} \mathcal{L}(\pi_\theta)$ has a unique optimal solution $\pi^*$ expressed in Equation (16) , and no other local or global minima exist.*

*Proof.* Let $J = \text{Supp}(\pi) = \{y_1, \ldots, y_n\}$, where $n = |J|$, and $\Pi$ be the set of policies with support set $J$. It is straightforward that $\min_{\pi \in \Pi} \mathcal{L}(\pi) = \mathcal{L}(\pi^*) = 0$, thus $\pi^*$ is a global optimal solution. We now prove the uniqueness of this optimal solution by the re-parameterization trick.

We parameterize $\Pi$ via vectors of logits $s \in \mathbb{R}^J$ of $\pi$, *i.e.*, $s_i = \log(\pi(y_i))$. Set $\pi_s(y) = \frac{\exp(s_i)}{\sum_{i=1}^n \exp(s_i)}$ for $y = y_i \in J$ and $\pi_s(y) = 0$ otherwise. Specially, let $s^*$ be the vector of logits corresponding to $\pi^*$, we have $\pi^* = \pi_{s^*}$.

We first prove that $s^*$ is the global optimal solution to the optimization problem

$$\mathcal{L}(s) := \mathcal{L}(\pi_s) = \mathbb{E}_{y,y' \sim \pi_s} \left[ h_{\pi_s}(y, y') - \frac{g(y) - g(y')}{\tau} \right]^2.$$

It is obvious that $\mathcal{L}(s^*) = 0$, thus it is a local minimum. By expanding the square term, we have

$$\mathcal{L}(s) = \mathbb{E}_{y,y' \sim \pi_s} \left[ \frac{g(y) - g(y')}{\tau} - (s(y) - s(y')) - \log\left( \frac{\pi_{\text{ref}}(y')}{\pi_{\text{ref}}(y)} \right) \right]^2$$

$$= \sum_{y,y' \in J} \pi_s(y)\pi_s(y') \left[ ((s(y) - s(y'))^2 + C_1 \cdot ((s(y) - s(y')) + C_2 \right],$$

where $C_1, C_2$ are two terms independent of $s$. The above equation is a positive semidefinite quadratic form, and hence is convex. Thus, all local minima are global minima.

Now we prove that $\pi_{s^*}$ is the unique global minima to $\mathcal{L}(s)$. Since $\pi_s$ is a surjective continuous mapping from $s$ to $\pi$, then every local minima $\pi$ to $\mathcal{L}(\pi)$ corresponds to a set of $s$ that minimizes $\mathcal{L}(s)$. The uniquess of $s^*$ will deduce that $\pi^*$ is the unique optimal solution to Equation (18) and concludes the proof. Consider $s' = s^* + r \cdot \Delta s$, where the only $r$ is the radius and $\Delta s$ is the direction under the polar coordinate. The only direction that not increase $\mathcal{L}(s')$ away from 0 is $e = (\frac{1}{n}, \ldots, \frac{1}{n})$ (Boyd & Vandenberghe (2004), Chap. 3). However, we have

$$\pi_{s^*+r \cdot e}(s_i) = \frac{\exp(s_i + r \cdot \frac{1}{n})}{\sum_{i=1}^n \exp(s_i + r \cdot \frac{1}{n})} = \frac{\exp(s_i)}{\sum_{i=1}^n \exp(s_i)} = \pi_{s^*}(s_i), \ \forall i \in [n].$$

This indicates that $\pi_{s^*}$ is the unique global minima to $\mathcal{L}(\pi_{s^*})$ and thus $\pi^*$ is the unique optimal solution to Equation (18). $\qquad \square$

Now we provide the proof of Theorem 3.1, most of which follows Azar et al. (2024).

*Proof.* Let $\mathcal{S}$ be the set of all possible token combinations with fixed token length, then it is finite. Let $f(s) = (p^*_{\text{safe}}(s) + E_s)(p^*_{\text{help}}(s \succ \pi) + \frac{1}{2})$, $\delta(s) = \pi_\theta(s)$ and $\eta(s) = \pi_{\text{ref}}(s)$. All the conditions in the Lemma B.2 are satisfied. Thus, Equation (16) is a solution to the optimization problem in Equation (17).

Now we prove Equation (16) is also a solution to the optimization problem Equation (18). Plug Equation (16) in the Equation (18), we have

$$h_{\pi^*}(y, y') = \log\left( \frac{\pi^*(y)\pi_{\text{ref}}(y')}{\pi^*(y')\pi_{\text{ref}}(y)} \right) = \log\left( \frac{\exp\left(\tau^{-1}g(y)\right)}{\exp\left(\tau^{-1}g(y')\right)} \right) = \tau^{-1}(g(y) - g(y')),$$

which validates Equation (16) is a solution to the optimization problem Equation (18).

Finally, Lemma B.3 indicates Equation (16) is the unique solution to Equation (18). This concludes the proof. $\qquad \square$

The above proof holds for any order of $y, y'$ since the equation in Equation (19) is skew-symmetric, *i.e.*,

$$\left[ h_\pi(y, y') - \frac{g(y) - g(y')}{\tau} \right]^2 = \left[ h_\pi(y', y) - \frac{g(y') - g(y)}{\tau} \right]^2.$$

This allows us to freely arrange the order of $y, y'$ in Equation (18) without loss of generality. Therefore, Equation (18) can be written as

$$\min_{\pi_\theta} \mathbb{E}_{y,y' \sim \pi_\theta(y)} \left[ h_\pi(y^{hw}, y^{hl}) - \frac{g(y^{hw}) - g(h^{hl})}{\tau} \right]^2,$$

where

$$y^{hw} = \begin{cases} y & \text{if } I_{\text{help}}(y \succ y'|x) = 1, \\ y' & \text{otherwise,} \end{cases}$$

and

$$y^{hl} = \begin{cases} y' & \text{if } I_{\text{help}}(y \succ y'|x) = 1, \\ y & \text{otherwise.} \end{cases}$$

With this reordering, the theorem reduces to Theorem 3.1

### B.3 PROOF OF THEOREM 3.2

In this section, we prove the Theorem 3.2. We begin by rewriting the formula in Equation (12) into a function of $y, y'$.

$$\begin{aligned} g_I(y, y') = B_3 \Big( B_1 \big( I_{\text{safe}}(y) I_{\text{help}}(y \succ y') + I_{\text{safe}}(y') I_{\text{help}}(y' \succ y) \big) \\ - \big( I_{\text{safe}}(y) I_{\text{help}}(y' \succ y) + I_{\text{safe}}(y') I_{\text{help}}(y \succ y') \big) + B_2 \Big) \cdot \Big( 2 I_{\text{help}}(y \succ y') - 1 \Big), \end{aligned} \tag{21}$$

Here, $I_{\text{help}}(y \succ y')$ determines whether $y$ is the win response or lose response. In other words,

$$I_{\text{safe}}(y^{hw}) = I_{\text{safe}}(y) I_{\text{help}}(y \succ y') + I_{\text{safe}}(y') I_{\text{help}}(y' \succ y),$$

and the same applies to $I_{\text{safe}}(y^{hl})$. To enable the reordering of the variables, we further multiply the formula by $2 I_{\text{help}}(y \succ y') - 1$, since $h_\pi(y, y') = -h_\pi(y', y)$ By organizing the terms, we have

$$\begin{aligned} g_I(y, y') =& (B_1 B_3 - B_3) I_{\text{help}}(y \succ y') I_{\text{safe}}(y) + (B_1 B_3 - B_3) I_{\text{help}}(y \succ y') I_{\text{safe}}(y') \\ & - B_1 B_3 I_{\text{safe}}(y') + B_3 I_{\text{safe}}(y) + 2 B_2 B_3 I_{\text{help}}(y \succ y') - B_2 B_3 \end{aligned}$$

We first establish the equivalence of the two optimization problems in Equation (22) and Equation (23) under the specific choice of constants, and then provide the general relation of constants for the equivalence.

Here, we use the following constants:

$$A_1 = E_s, A_2 = \frac{1}{2}, B_1 = 3, B_2 = 0, B_3 = \frac{1}{2}.$$

**Theorem B.4.** *The optimization problem*

$$\min_{\pi_\theta} \mathbb{E}_{x \sim \rho, y, y' \sim \pi_\theta(y)} \left[ h_\pi(y, y') - \frac{g\big(p^*_{safe}(y), p^*_{help}(y)\big) - g\big(p^*_{safe}(y'), p^*_{help}(y')\big)}{\tau} \right]^2, \tag{22}$$

*where $g(y) = (p^*_{safe}(y) + E_s)(p^*_{help}(y \succ \pi) + \frac{1}{2})$, is equivalent to the optimization problem*

$$\min_{\pi_\theta} \mathbb{E}_{x \sim \rho, y, y' \sim \pi_\theta(y), I \sim Bernoulli} \left[ \left( h_\pi(y, y') - \frac{g_I(y, y')}{\tau} \right)^2 \right], \tag{23}$$

*where*

$$g_I(y, y') = I_{help}(y \succ y') I_{safe}(y) + I_{help}(y \succ y') I_{safe}(y') + \frac{1}{2} I_{safe}(y) - \frac{3}{2} I_{safe}(y')$$

Here, $I \sim$ Bernoulli denotes the Bernoulli variables $I_{\text{safe}}(y)$ and $I_{\text{safe}}(y')$.

*Proof.* The two minimization problems are both over $\pi_\theta$, so we only need to focus on the terms that involve $\pi_\theta$. Specifically, the first term and the cross term after expanding the square expression in the two minimization problems. The first term is the same. Here we prove the cross term is also the same.

Let $\pi_y = \log(\pi(y)), \pi_y^R = \log(\pi_{\text{ref}}(y))$, then we can write

$$h_\pi(y, y') = \pi_y - \pi_{y'} + \pi_{y'}^R - \pi_y^R$$

Let $p_h(y) = p^*_{\text{help}}(y \succ \pi)$ and $p_s(y) = p^*_{\text{safe}}(y)$. The cross term of Equation (22) can be written as

$$\mathbb{E}_{x\sim\rho,y,y'\sim\pi}\left[h_\pi(y,y')\left(g\left(p^*_{\text{safe}}(y),p^*_{\text{help}}(y \succ \pi)\right) - g\left(p^*_{\text{safe}}(y'),p^*_{\text{help}}(y' \succ \pi)\right)\right)\right]$$

$$=\mathbb{E}_{x\sim\rho,y,y'\sim\pi}\left[(\pi_y - \pi_{y'} + \pi^R_{y'} - \pi^R_y)\left(g\left(p_s(y),p_h(y)\right) - g\left(p_s(y'),p_h(y')\right)\right)\right]$$

$$=\mathbb{E}_{x\sim\rho,y\sim\pi}\left[(\pi_y - \pi^R_y)\left(g(p_s(y),p_h(y)) - \mathbb{E}_{y'\sim\pi}\left[g\left(p_s(y'),p_h(y')\right)\right]\right)\right]$$

$$+\mathbb{E}_{x\sim\rho,y'\sim\pi}\left[(-\pi_{y'} + \pi^R_{y'})\left(\mathbb{E}_{y\sim\pi}\left[g(p_s(y),p_h(y))\right] - g(p_s(y'),p_h(y'))\right)\right]$$

(24)

The third equality is by the independence of $y$ and $y'$. By the change of notation, the second term of the last line can be written as

$$\mathbb{E}_{x\sim\rho,y'\sim\pi}\left[(-\pi_{y'} + \pi^R_{y'})\left(\mathbb{E}_{y\sim\pi}\left[g(p_s(y),p_h(y))\right] - g(p_s(y'),p_h(y'))\right)\right]$$

$$=\mathbb{E}_{x\sim\rho,y\sim\pi}\left[(-\pi_y + \pi^R_y)\left(\mathbb{E}_{y'\sim\pi}\left[g(p_s(y'),p_h(y'))\right] - g(p_s(y),p_h(y))\right)\right]$$

(25)

Then Equation (24) can be written as

$$(24) = \mathbb{E}_{x\sim\rho,y\sim\pi}\left[(\pi_y - \pi^R_y)\cdot 2\left(g(p_s(y),p_h(y)) - \mathbb{E}_{y'\sim\pi}\left[g(p_s(y'),p_h(y'))\right]\right)\right] \qquad (26)$$

Now we plug in $g(p_s(y),p_h(y)) = (p_s(y)+E_s)(p_h(y)+\frac{1}{2})$ and use the fact $\mathbb{E}_{y'\sim\pi}[p_h(y' \succ \pi)] = \frac{1}{2}$. Equation (26) can be expanded as

$$(24) = \mathbb{E}_{x\sim\rho,y\sim\pi}\left[(\pi_y - \pi^R_y)\cdot 2\left((p_s(y)+E_s)(p_h(y)+\frac{1}{2}) - \mathbb{E}_{y'\sim\pi}\left[(p_s(y')+E_s)(p_h(y')+\frac{1}{2})\right]\right)\right]$$

$$=\mathbb{E}_{x\sim\rho,y\sim\pi}\left[(\pi_y - \pi^R_y)\cdot 2\left((p_s(y)+E_s)(p_h(y)+\frac{1}{2}) - 2E_s\right)\right]$$

$$=\mathbb{E}_{x\sim\rho,y\sim\pi}\left[(\pi_y - \pi^R_y)\cdot(2p_s(y)p_h(y) + 2E_sp_h(y) + p_s(y) - 3E_s)\right]$$

(27)

The cross term of Equation (23) can be written as

$$\mathbb{E}_{x\sim\rho,y,y'\sim\pi}\mathbb{E}_{I\sim\text{Bernoulli}}\left[h_\pi(y,y')g_I(y,y')\right]$$

$$=\mathbb{E}_{x\sim\rho,y,y'\sim\pi}\mathbb{E}_{I\sim\text{Bernoulli}}\left[(\pi_y - \pi_{y'} + \pi^R_{y'} - \pi^R_y)g_I(y,y')\right]$$

(28)

Now we plug in $g_I = I_{\text{help}}(y \succ y')I_{\text{safe}}(y) + I_{\text{help}}(y \succ y')I_{\text{safe}}(y') + \frac{1}{2}I_{\text{safe}}(y) - \frac{3}{2}I_{\text{safe}}(y')$,

$$(28) = \mathbb{E}_{x\sim\rho,y,y'\sim\pi}\mathbb{E}_{I\sim\text{Bernoulli}}\Big[(\pi_y - \pi_{y'} + \pi^R_{y'} - \pi^R_y)\big(I_{\text{help}}(y \succ y')I_{\text{safe}}(y)$$

$$+ I_{\text{help}}(y \succ y')I_{\text{safe}}(y') + \frac{1}{2}I_{\text{safe}}(y) - \frac{3}{2}I_{\text{safe}}(y')\big)\Big]$$

$$=\mathbb{E}_{x\sim\rho,y,y'\sim\pi}\mathbb{E}_{I\sim\text{Bernoulli}}\Big[(\pi_y - \pi^R_y)\big(I_{\text{help}}(y \succ y')I_{\text{safe}}(y)$$

$$+ I_{\text{help}}(y \succ y')I_{\text{safe}}(y') + \frac{1}{2}I_{\text{safe}}(y) - \frac{3}{2}I_{\text{safe}}(y')\big)\Big]$$

$$+ \mathbb{E}_{x\sim\rho,y,y'\sim\pi}\mathbb{E}_{I\sim\text{Bernoulli}}\Big[(-\pi_{y'} + \pi^R_{y'})\big(I_{\text{help}}(y \succ y')I_{\text{safe}}(y)$$

$$+ I_{\text{help}}(y \succ y')I_{\text{safe}}(y') + \frac{1}{2}I_{\text{safe}}(y) - \frac{3}{2}I_{\text{safe}}(y')\big)\Big]$$

With the similar change of notation as Equation (25), as well as the fact that $1 - I_{\text{help}}(y \succ y') = I_{\text{help}}(y' \succ y)$, the last line can be written as

$$\mathbb{E}_{x\sim\rho,y,y'\sim\pi}\mathbb{E}_{I\sim\text{Bernoulli}}\Big[(-\pi_{y'} + \pi^R_{y'})\big(I_{\text{help}}(y \succ y')I_{\text{safe}}(y)$$

$$+ I_{\text{help}}(y \succ y')I_{\text{safe}}(y') + \frac{1}{2}I_{\text{safe}}(y) - \frac{3}{2}I_{\text{safe}}(y')\big)\Big]$$

$$=\mathbb{E}_{x\sim\rho,y,y'\sim\pi}\mathbb{E}_{I\sim\text{Bernoulli}}\Big[(-\pi_y + \pi^R_y)\big(I_{\text{help}}(y' \succ y)I_{\text{safe}}(y')$$

$$+ I_{\text{help}}(y' \succ y)I_{\text{safe}}(y) + \frac{1}{2}I_{\text{safe}}(y') - \frac{3}{2}I_{\text{safe}}(y)\big)\Big]$$

$$=\mathbb{E}_{x\sim\rho,y,y'\sim\pi}\mathbb{E}_{I\sim\text{Bernoulli}}\Big[(-\pi_y + \pi^R_y)\big((1 - I_{\text{help}}(y \succ y'))I_{\text{safe}}(y')$$

$$+ (1 - I_{\text{help}}(y \succ y'))I_{\text{safe}}(y) + \frac{1}{2}I_{\text{safe}}(y') - \frac{3}{2}I_{\text{safe}}(y)\big)\Big]$$

Then we further expand Equation (28) as

$$
\begin{aligned}
(28) =& \mathbb{E}_{x\sim\rho, y, y'\sim\pi}\mathbb{E}_{I\sim\text{Bernoulli}}\Big[(\pi_y - \pi_y^R)\big(I_{\text{help}}(y \succ y')I_{\text{safe}}(y) \\
& \qquad\qquad\qquad\qquad + I_{\text{help}}(y \succ y')I_{\text{safe}}(y') + \frac{1}{2}I_{\text{safe}}(y) - \frac{3}{2}I_{\text{safe}}(y'))\Big] \\
& + \mathbb{E}_{x\sim\rho, y, y'\sim\pi}\mathbb{E}_{I\sim\text{Bernoulli}}\Big[(-\pi_y + \pi_y^R)\big((1 - I_{\text{help}}(y \succ y'))I_{\text{safe}}(y') \\
& \qquad\qquad\qquad\qquad + (1 - I_{\text{help}}(y \succ y'))I_{\text{safe}}(y) + \frac{1}{2}I_{\text{safe}}(y') - \frac{3}{2}I_{\text{safe}}(y))\Big] \\
=& \mathbb{E}_{x\sim\rho, y, y'\sim\pi}\mathbb{E}_{I\sim\text{Bernoulli}}\Big[(\pi_y - \pi_y^R)\big(2I_{\text{help}}(y \succ y')I_{\text{safe}}(y) \\
& \qquad\qquad\qquad\qquad + 2I_{\text{help}}(y \succ y')I_{\text{safe}}(y') + I_{\text{safe}}(y) - 3I_{\text{safe}}(y'))\Big]
\end{aligned}
\tag{29}
$$

Taking the expectation over $y'$ and the Bernoulli variables, we have

$$
(28) = \mathbb{E}_{x\sim\rho, y\sim\pi}\Big[(\pi_y - \pi_y^R)\big(2p_h(y)p_s(y) + 2E_s p_h(y) + p_s(y) - 3E_s\big)\Big]
\tag{30}
$$

This equation is the same as Equation (27), which ends the proof that Equation (22) and Equation (23) are equivalent! $\qquad\square$

As discussed in Appendix B.2, we can freely change the order of $y$ and $y'$ in Equation (22) and Equation (23). Thus, the proof of Theorem B.4 also applies to Theorem 3.2.

## B.4 RELATION OF THE CONSTANTS

In this section, we derive a more general form of Theorem B.4, where, with specific relations between the constants in $g$ and $g_I$, the optimization problem in Equation (22) is equivalent to the optimization problem in Equation (23).

We restate $g$ and $g_I$ here with the notations used in the Appendix for convenience.

$$
g = (p_s(y) + A_1)(p_h(y) + A_2),
$$

and

$$
\begin{aligned}
g_I(y, y') =& (B_1 B_3 - B_3)I_{\text{help}}(y \succ y')I_{\text{safe}}(y) + (B_1 B_3 - B_3)I_{\text{help}}(y \succ y')I_{\text{safe}}(y') \\
& - B_1 B_3 I_{\text{safe}}(y') + B_3 I_{\text{safe}}(y) + 2B_2 B_3 I_{\text{help}}(y \succ y') - B_2 B_3
\end{aligned}
$$

As discussed in the proof of Theorem B.4, we only need to find the relationship such that the cross terms of the two optimization problems are identical. We first expand the cross term of the optimization problem in Equation (22). As in Equation (26), it can be written as

$$
(24) = \mathbb{E}_{x\sim\rho, y\sim\pi}\Big[(\pi_y - \pi_y^R) \cdot 2\big(g(p_s(y), p_h(y)) - \mathbb{E}_{y'\sim\pi}[g(p_s(y'), p_h(y'))]\big)\Big]
\tag{31}
$$

Using the same strategy of obtaining Equation (29), we have

$$
\begin{aligned}
(28) = \mathbb{E}_{x\sim\rho, y\sim\pi}\Big[& (\pi_y - \pi_y^R)\big(2B_3(B_1 - 1)p_s(y)p_h(y) \\
& + 2B_3((B_1 - 1)E_s + 2B_2)p_h(y) + 2B_3 p_s(y) - 2B_1 B_3 E_s - 2B_2 B_3\big)\Big]
\end{aligned}
\tag{32}
$$

Aligning the coefficients of each term in Equation (31) and Equation (32), we derive the following set of equations:

$$
\begin{aligned}
B_3(B_1 - 1) &= 1, \\
E_s + 2B_3 B_3 &= A_1, \\
B_3 &= A_2.
\end{aligned}
\tag{33}
$$

Solving these equations gives us the specific forms of $g$ and $g_I$. Here $B_2$ is a shifting value that we define to align with our intuition. $B_3$ is a scaling factor that is related to the penalty $\tau$.

## B.5 Discussion of the Property of $g_I$

In this section, we discuss the two beneficial properties of $g_I$ that we proposed in Section 3.2.

**Skew-Symmetric Property.** First, we examine the skew-symmetric property of $g_I$. When combined with the skew-symmetric property of $h$, this implies:

$$\left(h_\pi(y, y') - \tau^{-1}g_I(y, y')\right)^2 = \left(h_\pi(y', y) - \tau^{-1}g_I(y', y)\right)^2.$$

This means that for the same data point, regardless of the order of $y$ and $y'$, we are always driving $h_\pi(y, y')$ to the same value. In contrast, in IPO (Azar et al., 2024), different orders will push $h_\pi(y, y')$ to different values, i.e., they form two different optimization problems:

$$(h_\pi(y, y') - \tau^{-1}g_I(y, y'))^2 \quad \text{and} \quad (h_\pi(y', y))^2.$$

Their final optimization problem, $(h_\pi(y, y') - \frac{1}{2}\tau^{-1}g_I(y, y'))^2$, tries to find a middle point of $h$ that optimizes both. However, this point is neither the optimal solution of the first problem nor the second problem.

**Shifting Property.** Second, we discuss the shifting properties of $g_I$. Since Theorem 3.2 holds based on the equality of Equation (30) and Equation (27), and all the operations to derive these two equations are valid under linear transformations of $p^*_{\text{safe}}, p^*_{\text{help}}$ and $I_{\text{safe}}, I_{\text{help}}$, respectively. It implies that Theorem 3.2 also holds under the same linear transformations of $p^*_{\text{safe}}, p^*_{\text{help}}$ and $I_{\text{safe}}, I_{\text{help}}$.

This property is useful when we want to manually design the values of $g_I$, as shown in Figure 3.

## C Experiment

### C.1 Details of Harmlessness Benchmark

The following are the details of the datasets used in the benchmark:

- **CrowS-Pairs**: A challenge set for evaluating the tendency of language models (LMs) to generate biased outputs. We evaluate the English subset and implementation by LM Evaluation Harness and report the Percentage of Stereotype metric.

- **Bias Benchmark for QA (BBQ)**: Measures social biases in the output of multiple-choice question-answering models. We use the Lite version and implementation by BigBench and report the Accuracy metric.

- **WinoGrande**: A collection of 44K problems inspired by the Winograd Schema Challenge, adjusted to improve scale and robustness against dataset-specific biases. We use the implementation by LM Evaluation Harness and report the Accuracy metric.

- **ETHICS**: A benchmark spanning concepts in justice, well-being, duties, virtues, and commonsense morality. We use the implementation by LM Evaluation Harness and report the Accuracy for all sub-tasks.

- **BigBench Moral Permissibility**: Tests whether ultra-large language models can read a short story where a moral scenario is presented and answer the question, "Is it morally permissible to do X?" in a manner similar to humans. We use the implementation by BigBench and report the Accuracy metric.

- **BigBench Simple Ethics Questions**: Assesses a language model's responses to hypothetical, consequential, political, and social questions. We use the implementation by BigBench and report the Accuracy metric.

- **ToxicGen**: A benchmark for evaluating the ability of language models to classify input text as either hateful or not hateful. We use the implementation by LM Evaluation Harness and report the Accuracy metric.

- **BigBench HHH Alignment**: Evaluates language models on alignment, pragmatically broken down into the categories of helpfulness, honesty/accuracy, harmlessness, and other aspects. We use the implementation by BigBench and report the Accuracy metric.

- **AdvBench** contains harmful prompts. We use the prompts provided here and generation implementation by LM Evaluation Harness. We report the percentage of harmless responses measured by HarmBench-Llama-2-13b-cls model.

- **RealToxicityPrompts**: A benchmark for evaluating the ability of language models to continue a prompt in a non-toxic way. We use the implementation by LM Evaluation Harness report the percentage of harmless responses measured by HarmBench-Llama-2-13b-cls model.

- **ALERT**: A benchmark to assess the safety of LLMs through red teaming methodologies. We use the prompts provided here and generation implementation by LM Evaluation Harness. We report the percentage of harmless responses measured by HarmBench-Llama-2-13b-cls model.

- **ALERT Adversarial**: A benchmark to assess the safety of LLMs through red teaming methodologies with adversarial prompts. We use the prompts provided here and generation implementation by LM Evaluation Harness. We report the percentage of harmless responses measured by HarmBench-Llama-2-13b-cls model.

- **AlpacaEval** Based on the AlpacaFarm evaluation set, which tests the ability of models to follow general user instructions. We employ the official implementation report the LC Win Rate.

## C.2 DETAILS OF BASELINES

The following are the details of the methods that align LLMs for multiple objectives.

- **Llama2** (Touvron et al., 2023) trains the safety reward $r_{\text{safe}}$ and the helpfulness reward $r_{\text{help}}$ separately, and defines the global reward $g$ as a combination of these rewards, *i.e.*,

$$\tilde{g}(y|x) = \begin{cases} r_{\text{safe}}(y|x) \text{ if IS\_SAFETY}(x), \text{ or } r_{\text{safe}}(y|x) < 0.15, \\ r_{\text{help}}(y|x) \text{ otherwise}, \end{cases}$$

$$g(y|x) = \text{WHITEN}(\text{LOGIT}(\tilde{g}(y|x))).$$

Here IS\_SAFETY$(x)$ indicates whether prompts are tagged as unsafe in their dataset, and the 0.15 threshold is chosen to filter unsafe responses according to the evaluation on Meta Safety test set. Whitening the final linear scores is to increase stability. The global reward is used in the RLHF objective in Equation (3).

- **Beaver** (Dai et al., 2024) trains the safety reward $r_{\text{safe}}$ and the helpfulness reward $r_{\text{help}}$ separately, and defines the final RLHF objective as the dual optimization problem of the conditional RLHF, obtained by Lagrangian dual transformation, *i.e.*,

$$\min_{\theta} \max_{\lambda \geq 0} \mathbb{E}_{x \sim D, y \sim \pi_{\theta}} \left[ -r_{\text{help}}(y|x) + \lambda \left( r_{\text{safe}}(y|x) + d \right) \right],$$

where $\lambda \geq 0$ is the Lagrange multiplier. In practice, the model parameter $\theta$ and the Lagrange multiplier $\lambda$ are updated iteratively.

- **RBR** (Mu et al., 2024) requires separate reward models, $r_{\phi_1}, \dots, r_{\phi_k}$, for each objective, and propose to learn the weight for each objective, *i.e.*,

-

$$g(y|x) = \sum_{i=1}^{k} \lambda_i r_i(y|x),$$

where $\lambda_i$ are learnable parameters. The global reward is used in the RLHF objective in Equation (3).

- **SteerLM** (Dong et al., 2023) trains models to generate response according to a specific reward vector $r = (r_1, r_2, r_3, \dots, r_k)$. They first train a model to predict the score for each objective in a dataset. Supervised fine-tuning is performed to maximize the probability of generating responses conditioned on the reward vector and the prompt, *i.e.*,

$$\max_{\theta} \mathbb{E}_{(x,y,r) \sim D} \log p_{\theta}(y|x, r).$$

- **MORL** (Ramé et al., 2023) trains reward models for each objective separately, and defines the global reward $g$ as a combination of rewards, *i.e.*,

$$g(y|x) = \sum_{i=1}^{k} \lambda_i r_i(y|x),$$

  The global reward is used in the RLHF objective in Equation (3).

- **ArmoRM** (Wang et al., 2024b) applies the same training strategy as MORL, but uses a single publicly available reward model, ArmoRM-Llama3-8B-v0.1 (Wang et al., 2024c), to provide the reward scores for all objectives.

- **MODPO** (Zhou et al., 2023) trains margin reward models $r_i, i = 1, \ldots, k$ for each objective separately, and performs supervised fine-tuning with the objective,

$$\max \mathbb{E}_{(x,y^w,y^l) \sim D}$$
$$\log \sigma \left( \frac{1}{\omega_k} \left( \tau \log \frac{\pi_\theta(y^w|x)}{\pi_{\text{ref}}(y^w|x)} - \tau \log \frac{\pi_\theta(y^l|x)}{\pi_{\text{ref}}(y^l|x)} - \omega_{-k}^T(r_{-k}(x, y^w) - r_{-k}(x, y^l)) \right) \right),$$

  where $\omega_k$ is the weight for the objective $k$, $\omega_{-k}$ is the weight vector for all other objectives, and $r_{-k}$ is the reward vector for all other objectives than $k$. This fine-tuning is performed for each objective.

- **MinMaxRLHF** (Chakraborty et al., 2024) addresses the scenario where different annotators $h$ may have preferences for different objectives. The algorithm uses the EM algorithm to learn the distribution of rewards for multiple objectives. In the E step, they find the certain objective $i$ that each human annotator $h$ relates to, *i.e.*,

$$\mathcal{I}_h = \arg \max_i \Pi_{x,y,y',h} \frac{\exp(r_{\phi_i}(x, y))}{\exp(r_{\phi_i}(x, y)) + \exp(r_{\phi_i}(x, y'))},$$

  where $r_{\phi_i}$ is the reward model for the objective $i$. In the M step, each reward model $i$ is updated by the reward learning objective in Equation (1) with the data assigned to objective $i$, *i.e.*, the dataset is $D_i = \{(x, y, y', h), \mathcal{I}_h = i\}$. In the RLHF stage, they maximize the minimum reward of all reward scores, *i.e.*,

$$\mathbb{E}_{x \sim D, y \sim \pi_\theta} \left[ \min_i r_{\phi_i}(x, y) - \tau \text{KL}\left[ \pi_\theta(y|x) || \pi_{\text{ref}}(y|x) \right] \right].$$

Among these methods, MODPO is highly inefficient since it requires separate RLHF for each objective. Other methods typically use a linear combination of reward scores for multiple objectives or one reward as a threshold for others. For the combination of thresholding, the global function can be approximated by the multiplication of rewards for each objective when the reward scores are on the same scale. Maximizing the multiplication of rewards has the same effect as maximizing the minimum reward. Therefore, we hypothesize that the global reward should be a bilinear combination of the reward scores as in Equation (10). As such, in the experiment section, we select MORL as a representative for this line of approach.

## C.3 ADDITIONAL ABLATION STUDIES

We include additional ablation studies of other hyper-parameters in our algorithms. Full results are in Table 10.

**LoRA finetuning.** We follow the setting of Section 4.2 and conduct additional experiments to compare our method and the best performed baseline, IPO, with LoRA fine-tuning. During the training, we apply the same training hyper-parameters to both algorithms, like learning rate, training epochs, beta, and so on. The results are in Table 6. Results show that BFPO consistently outperformed the baselines when training with LoRA. However, we observed that LoRA training required additional hyperparameter tuning, which posed challenges due to the limited time. Consequently, both methods achieved lower overall performance and worse balance compared to selective fine-tuning.

| | LoRA Fine-tuning | | | | Selective Fine-tuning | | | |
|---|---|---|---|---|---|---|---|---|
| | Helpfulness | Harmlessness | | | Helpfulness | Harmlessness | | |
| | Alpaca | Disc. | Gen. | Savg. | Alpaca | Disc. | Gen. | Savg. |
| IPO | 6.14 | 58.05 | 93.97 | 76.1 | 13.15 | 58.41 | 89.76 | 74.09 |
| BFPO | 7.77 | 64.36 | 94.73 | 79.54 | 3.33 | 59.09 | 95.24 | 77.16 |

Table 6: Comparison of Helpfulness and Harmlessness Metrics with LoRA finetuning

Table 7: Ablation Study of $\alpha$ in Equation (15)

| $\alpha$ | Helpfulness | Harmlessness | | |
|---|---|---|---|---|
| | Alpaca | Disc. | Gen. | Savg. |
| 0.1 | 13.61 | 59.81 | 87.39 | 73.60 |
| 0.3 | 14.06 | 60.31 | 91.73 | 76.02 |
| 0.5 | 13.33 | 59.09 | 95.24 | 77.16 |
| 0.7 | 9.01 | 57.34 | 96.28 | 76.81 |
| 0.9 | 7.21 | 56.44 | 96.66 | 76.55 |

Table 8: Ablation Study of $\tau$ in Equation (15)

| $\tau$ | Helpfulness | Harmlessness | | |
|---|---|---|---|---|
| | Alpaca | Disc. | Gen. | Savg. |
| 0.01 | 13.33 | 59.09 | 95.24 | 77.16 |
| 0.1 | 6.4 | 55.44 | 81.45 | 68.44 |
| 0.5 | 6.53 | 54.01 | 78.14 | 66.07 |
| 1.0 | 6.74 | 54.10 | 77.52 | 65.81 |

**The hyperparameter $\alpha$.** The hyperparameter controls the label values (represent the difference of the preference of a pair of response) of the four cases in Figure 3. To ensure the desired behavior, that helpful-safe responses are preferred over helpless-safe ones (case 2 in Figure 3 yields a positive value) and that helpful-unsafe responses are not preferred over helpless-unsafe ones (Case 3 in Figure 3 yields a negative value), we constrain $\alpha \in (0, 1)$. When $\alpha = 0.5$, the label values for the four cases are $1, 0.5, -0.5, -1$, where the absolute label values are symmetric for positive and negative pairs. As $\alpha$ increases, the absolute label values in case 1,2 in Figure 3 decrease, and the absolute label values in case 3,4 in Figure 3 increase. In other words, positive pairs will have smaller differences and negative pairs will have larger differences.

In the ablation study, we follow the experiment of Section 4.2 with values of $0.1, 0.3, 00.5, 0.7, 0.9$ to systematically explore its effects. The results in Table 7 show that higher $\alpha$ values reduce distinctions between positive pairs, particularly helpful-safe vs. non-helpful-safe, leading to a lower helpfulness score. However, it increases distinctions between negative pairs, especially helpful-unsafe vs. non-helpful-safe, resulting in improved harmlessness, particularly in generative tasks.

**The hyperparameter $\tau$.** The hyperparameter $\tau$ is the coefficient of the KL term in Equation (7), which prevents the policy from deviating from the reference policy. In practice, it is important to note that is more related to the training and convergence (as shown in Figure 4) rather than being a core component to balance the safety and helpfulness.

In our experiments, we follow the settings from Tunstall et al. (2023), where $\tau = 0.01$ is used. This value is applied consistently across all baselines to ensure a fair comparison. For the ablation study, we adopt values inspired by Tang et al. (2024), specifically $\tau = 0.01, 0.1, 0.5, 1.0$. The results in Table 8 indicate that performance can vary significantly with different $\tau$ values. With different $\tau$, other training hyperparameters, like learning rate, training iterations also need to be carefully chosen.

**Ablation on $B_1, B_2, B_3$.** both $B_1, B_3$ must be positive. For this ablation study, we explore the following values $B_3 = 2, 1, \frac{1}{2}, \frac{1}{4}$. Given the constraint $B_3(B_1 - 1) = 1$ as in Equation (33), the corresponding values of $B_1$ are determined for each $B_3$. Additionally, $B_2$ is adjusted to balance the cases described in Figure 3 (Case 2 and Case 3). The experiment results are in Table 9.

When $B_3$ is smaller, the label differences for cases 1,2 and 3,4 in Figure 3 become less pronounced. For example, in Cases 1 and 2, the pairs (helpful-safe, non-helpful-unsafe) and (helpful-safe, non-helpful-safe) have smaller differences in their label values. This means there is less distinction in whether the non-helpful response is safe or not. As a result, the model shows slightly worse performance in helpfulness but performs better in safety. When $B_3$ is larger, the label differences for the aforementioned two cases become more distinct, and the label value for (helpful-safe, non-

helpful-unsafe) becomes significantly higher. This leads the model to prioritize safety more strongly, which results in improved safety performance but a sacrifice in helpfulness.

To conclude, larger $B_3$ values emphasize safety at the expense of helpfulness, while proper values allow for more balanced performance across both objectives.

Table 9: Ablation study for $B_1, B_2, B_3$ in Equation (9)

| $B_3$ | $B_1$ | $B_2$ | Values for four cases in Figure 3 | Helpfulness Alpaca | Harmlessness Disc. | Gen. | Savg. |
|---|---|---|---|---|---|---|---|
| 2 | 1.5 | -0.25 | 2.5,0.5,-0.5,-0.25 | 9.00 | 58.67 | 95.47 | 77.07 |
| 1 | 2 | -0.5 | 1.5,0.5,-0.5,-1.5 | 11.36 | 60.28 | 95.12 | 77.70 |
| 0.5 | 3 | -1 | 1,0.5,-0.5,-1 | 13.33 | 59.09 | 95.24 | 77.16 |
| 0.25 | 5 | -2 | 0.75, 0.5, -0.5, -0.75 | 13.15 | 59.63 | 94.25 | 76.94 |

## C.4 FULL EXPERIMENT RESULTS

Here are the details of each open-sourced models:

- Zephyr: `https://huggingface.co/HuggingFaceH4/zephyr-7b-beta`
- Juanako: `https://huggingface.co/fblgit/juanako-7b-UNA`
- OpenChat: `https://huggingface.co/openchat/openchat_3.5`
- Mistral: `https://huggingface.co/mistralai/Mistral-7B-Instruct-v0.2`
- Beaver: `https://huggingface.co/PKU-Alignment/beaver-7b-v3.0`
- Llama2: `https://huggingface.co/meta-llama/Llama-2-7b-chat-hf`
- Llama3: `https://huggingface.co/meta-llama/Meta-Llama-3-8B-Instruct`

Table 10 shows the full results of all experiments throughout the paper. Here are the details of the data used in our model and the baselines.

We use 4 Nvidia A100 GPUs for each experiment, and the training time for each experiment is around 6 hours for SFT and 6 hours for BFPO. For the experiments with red teaming data, we use 1.5K data collected as described in Section 4.3 and only performs the BFPO stage. The training time for this experiment is around 10 minutes with 4 Nvidia A100 GPUs.

Table 10: Full results of all experiments

| Model / Method | Alpaca Eval | Crows Pairs | BBQ | Winogrand | Ethics CM | Ethics Justice | Ethics Deontology | Ethics Utilitarianism | Ethics Virtue | Moral Permissibility | Simple Ethical Questions | Toxigen | HHH Alignment | Real-ToxicityPrompts | AdvBench | ALERT | ALERT Adversarial | Discriminative Average | Generative Average | Safety Average |
|---|---|---|---|---|---|---|---|---|---|---|---|---|---|---|---|---|---|---|---|---|
| Zephyr-7b-beta | 10.99 | 62.02 | 39.00 | 72.38 | 68.37 | 69.71 | 56.98 | 73.59 | 91.30 | 51.00 | 33.00 | 45.21 | 46.00 | 85.82 | 20.19 | 79.08 | 66.68 | 59.05 | 62.94 | 60.99 |
| Juanako-7b-UNA | 2.88 | 63.74 | 84.00 | 77.43 | 75.96 | 76.41 | 64.10 | 73.79 | 89.13 | 49.00 | 82.00 | 60.96 | 49.00 | 85.90 | 27.50 | 80.70 | 72.79 | 70.46 | 66.72 | 68.59 |
| OpenChatv3.5 | 11.08 | 66.67 | 61.00 | 72.69 | 68.88 | 77.74 | 63.96 | 73.48 | 88.70 | 50.00 | 91.00 | 42.34 | 46.00 | 87.82 | 48.27 | 75.36 | 73.09 | 66.87 | 71.13 | 69.00 |
| Mistral-7B-Instruct-v0.2 | 14.72 | 64.88 | 61.84 | 73.80 | 73.46 | 71.93 | 60.26 | 66.78 | 90.87 | 47.95 | 53.91 | 55.11 | 47.06 | 83.74 | 65.38 | 90.44 | 77.71 | 63.99 | 79.32 | 71.65 |
| Beaver3 | 1.00 | 56.23 | 31.37 | 65.35 | 59.43 | 64.61 | 61.48 | 56.01 | 61.61 | 47.66 | 45.22 | 36.17 | 43.44 | 85.07 | 93.20 | 91.83 | 94.80 | 52.38 | 91.23 | 71.80 |
| Llama2 | 7.60 | 63.98 | 32.99 | 66.46 | 56.14 | 50.00 | 50.00 | 57.97 | 72.00 | 47.37 | 24.35 | 51.00 | 44.34 | 87.91 | 100.00 | 98.62 | 98.32 | 51.38 | 96.21 | 73.80 |
| Llama3 | 22.90 | 63.45 | 60.68 | 71.82 | 58.64 | 70.38 | 64.49 | 62.92 | 81.49 | 48.54 | 54.78 | 45.74 | 45.25 | 89.49 | 99.42 | 95.18 | 95.08 | 60.68 | 94.79 | 77.74 |
| Mistral + DPO + Helpful Data | 10.99 | 62.02 | 39.00 | 72.38 | 68.37 | 69.71 | 56.98 | 73.59 | 91.30 | 51.00 | 33.00 | 45.21 | 46.00 | 85.82 | 20.19 | 79.08 | 66.68 | 59.05 | 62.94 | 60.99 |
| Mistral + DPO + Safety Data | 4.34 | 65.65 | 39.50 | 74.03 | 64.22 | 55.29 | 50.86 | 60.00 | 89.73 | 46.78 | 38.26 | 47.45 | 45.25 | 87.74 | 100.00 | 99.91 | 99.98 | 56.42 | 96.91 | 76.66 |
| Mistral + DPO + Naive mix Data | 14.71 | 65.59 | 43.68 | 74.27 | 56.47 | 71.01 | 58.20 | 57.15 | 86.71 | 51.17 | 39.13 | 51.06 | 45.70 | 82.49 | 4.23 | 38.64 | 33.46 | 58.35 | 39.71 | 49.03 |
| Mistral + IPO + Naive mix Data | 13.16 | 66.25 | 42.44 | 74.66 | 62.03 | 66.35 | 54.67 | 67.03 | 89.17 | 47.37 | 37.39 | 48.72 | 44.80 | 86.41 | 88.65 | 96.00 | 88.00 | 58.41 | 89.76 | 74.09 |
| Mistral + MORL + Naive mix Data | 10.83 | 61.66 | 39.43 | 71.51 | 68.01 | 67.71 | 55.70 | 72.57 | 91.08 | 50.58 | 33.91 | 44.15 | 46.15 | 87.07 | 21.15 | 82.13 | 69.16 | 58.54 | 64.88 | 61.71 |
| Mistral + BFPO + mix Data | 13.33 | 65.77 | 60.68 | 74.98 | 65.25 | 59.13 | 51.97 | 70.36 | 90.41 | 47.37 | 39.13 | 54.15 | 45.25 | 87.32 | 98.65 | 98.56 | 96.42 | 59.09 | 95.24 | 77.16 |
| Red Teaming + DPO | 13.07 | 61.84 | 38.95 | 72.45 | 67.77 | 69.12 | 57.48 | 73.63 | 91.42 | 50.88 | 36.52 | 45.11 | 46.15 | 87.99 | 44.00 | 89.22 | 76.33 | 59.28 | 74.39 | 66.83 |
| Red Teaming + IPO | 13.74 | 61.96 | 38.89 | 72.77 | 68.03 | 69.30 | 57.62 | 73.54 | 91.48 | 50.58 | 36.52 | 45.43 | 45.70 | 87.91 | 41.35 | 87.48 | 74.55 | 59.32 | 72.82 | 66.07 |
| Red Teaming + MORL | 12.56 | 61.66 | 38.47 | 71.98 | 66.07 | 69.16 | 56.98 | 73.02 | 91.36 | 51.17 | 33.04 | 44.26 | 45.70 | 87.66 | 21.15 | 82.13 | 69.16 | 58.57 | 65.02 | 61.80 |
| Red Teaming + BFPO | 14.41 | 61.72 | 39.44 | 72.45 | 67.28 | 68.01 | 57.20 | 73.46 | 91.54 | 49.12 | 37.39 | 45.32 | 45.25 | 87.82 | 86.54 | 86.34 | 94.47 | 59.02 | 88.79 | 73.90 |
| BFPO w/o shift | 12.76 | 65.95 | 44.44 | 74.98 | 62.50 | 59.87 | 52.28 | 66.64 | 88.78 | 47.66 | 50.43 | 49.89 | 45.70 | 84.32 | 96.15 | 96.90 | 94.12 | 59.09 | 92.87 | 75.98 |
| BFPO w/o buffer | 15.59 | 65.65 | 44.43 | 74.43 | 61.13 | 69.05 | 56.56 | 67.08 | 89.79 | 47.37 | 46.96 | 53.94 | 45.25 | 84.82 | 85.77 | 95.37 | 89.08 | 60.14 | 88.76 | 74.45 |
| IPO + LoRA Finetuning | 6.14 | 65.95 | 43.13 | 75.06 | 63.81 | 58.91 | 52.25 | 68.78 | 88.76 | 47.37 | 37.39 | 49.89 | 45.25 | 84.40 | 98.65 | 97.76 | 95.07 | 58.05 | 93.97 | 76.01 |
| BFPO + LoRA Finetuning | 7.77 | 66.25 | 46.41 | 74.90 | 72.20 | 73.00 | 58.40 | 66.53 | 91.38 | 47.08 | 64.35 | 66.06 | 45.70 | 85.82 | 98.85 | 98.23 | 96.03 | 64.36 | 94.73 | 79.54 |
| BFPO $\alpha = 0.1$ | 13.61 | 65.59 | 42.91 | 74.27 | 62.52 | 68.16 | 55.73 | 66.87 | 90.29 | 47.95 | 44.35 | 53.40 | 45.70 | 83.49 | 82.31 | 94.96 | 88.81 | 59.81 | 87.39 | 73.60 |
| BFPO $\alpha = 0.3$ | 14.06 | 65.41 | 43.71 | 74.82 | 62.50 | 67.01 | 55.76 | 68.34 | 90.19 | 47.66 | 50.43 | 52.13 | 45.70 | 84.82 | 93.46 | 96.61 | 92.05 | 60.31 | 91.73 | 76.02 |
| BFPO $\alpha = 0.5$ | 13.33 | 65.77 | 45.25 | 74.98 | 65.25 | 59.13 | 51.97 | 70.36 | 90.41 | 47.37 | 39.13 | 54.15 | 45.25 | 87.32 | 98.65 | 98.56 | 96.42 | 59.09 | 95.24 | 77.16 |
| BFPO $\alpha = 0.7$ | 9.01 | 65.71 | 43.06 | 75.06 | 68.34 | 54.33 | 51.22 | 67.74 | 89.97 | 47.08 | 26.96 | 53.83 | 44.80 | 86.57 | 99.81 | 99.72 | 99.01 | 57.34 | 96.28 | 76.81 |
| BFPO $\alpha = 0.9$ | 7.21 | 65.83 | 44.44 | 75.30 | 69.03 | 51.48 | 50.44 | 68.41 | 89.25 | 47.08 | 21.74 | 49.47 | 44.80 | 87.49 | 99.81 | 99.84 | 99.50 | 56.44 | 96.66 | 76.55 |
| BFPO $\tau = 0.01$ | 13.33 | 65.77 | 45.25 | 74.98 | 65.25 | 59.13 | 51.97 | 70.36 | 90.41 | 47.37 | 39.13 | 54.15 | 45.25 | 87.32 | 98.65 | 98.56 | 96.42 | 59.09 | 95.24 | 77.16 |
| BFPO $\tau = 0.1$ | 6.4 | 66.01 | 40.64 | 74.82 | 63.78 | 57.10 | 51.78 | 63.66 | 88.50 | 47.37 | 22.61 | 45.11 | 43.89 | 84.82 | 63.65 | 91.01 | 86.31 | 55.44 | 81.45 | 68.44 |
| BFPO $\tau = 0.5$ | 6.53 | 66.31 | 38.78 | 74.74 | 62.08 | 54.29 | 51.17 | 60.69 | 86.37 | 48.54 | 17.39 | 43.40 | 44.34 | 84.90 | 50.00 | 91.54 | 86.10 | 54.01 | 78.14 | 66.07 |
| BFPO $\tau = 1.0$ | 6.74 | 66.01 | 39.34 | 74.90 | 62.08 | 55.36 | 51.64 | 60.17 | 85.87 | 49.12 | 17.39 | 43.40 | 43.89 | 85.07 | 48.08 | 91.04 | 85.91 | 54.10 | 77.52 | 65.81 |
| BFPO $B_3 = 2$ | 9.00 | 65.65 | 45.72 | 75.61 | 70.35 | 52.59 | 50.61 | 68.51 | 89.73 | 46.78 | 38.26 | 54.57 | 45.70 | 85.49 | 99.62 | 98.92 | 97.85 | 58.67 | 95.47 | 77.07 |
| BFPO $B_3 = 1$ | 11.36 | 65.65 | 44.00 | 74.51 | 65.97 | 65.27 | 54.51 | 70.61 | 90.39 | 47.37 | 46.09 | 53.72 | 45.25 | 85.65 | 99.62 | 98.93 | 96.30 | 60.28 | 95.12 | 77.70 |
| BFPO $B_3 = 0.5$ | 13.33 | 65.77 | 45.25 | 74.98 | 65.25 | 59.13 | 51.97 | 70.36 | 90.41 | 47.37 | 39.13 | 54.15 | 45.25 | 87.32 | 98.65 | 98.56 | 96.42 | 59.09 | 95.24 | 77.16 |
| BFPO $B_3 = 0.25$ | 13.15 | 65.59 | 43.96 | 75.06 | 62.88 | 65.46 | 54.59 | 68.64 | 89.83 | 47.66 | 43.48 | 53.19 | 45.25 | 86.07 | 98.46 | 97.85 | 94.62 | 59.63 | 94.25 | 76.94 |

