# OpenReview forum: "Bi-Factorial Preference Optimization: Balancing Safety-Helpfulness in Language Models"
_ICLR.cc/2025/Conference — ICLR 2025 Spotlight_

### Official Review · Reviewer_Tsy7 · 2024-10-28

**Soundness:** 4
**Presentation:** 3
**Contribution:** 3
**Rating:** 8
**Confidence:** 3

**Summary:**

This paper propose Bi-Factorial Preference Optimization (BFPO) to learn a single objective of both safety and helpfulness. Specifically, the authors introduce a novel label function that scores preference in terms of both safety and helpfulness, and theoretically prove that solving a supervised optimization problem with the label function is equivalent to solving the multi-objective RLHF with a combination of the rewards of safety and helpfulness. Experimental result shows that BFPO achieves the highest harmlessness score and the best balance between helpfulness and harmlessness.

**Strengths:**

1. This paper propose a novel approach to align LLMs in terms of both harmlessness and helpfulness, which is essential for AI safety.
2. The proposed label function is theoretically equivalent to the multi-objective RLHF, possessing several properties that offer flexibility when constructing algorithms.
3. The authors conduct extensive experiments to strengthen the soundness of BFPO.

**Weaknesses:**

1. Some notations are inconsistence or not defined clearly, causing confusions in comprehend Sec. 2 and 3. For example, the reward function is written as $r(x,y)$ in some cases while $r(y|x)$ in others. Similarly, there are $g(x,y)$ and $g(y|x)$. Also, in eq 10 and 11, $y\succ \pi$ is not clearly defined in the main paper. The authors could choose one consistent notation (e.g., $r(y|x)$) and use it throughout the paper, and add a notation table to clarify the meaning of each symbol, including $y\succ \pi$.

**Questions:**

1. Appendix B.4 only shows that the relationship between $B_1$ and $B_3$ is $B_3(B_1-1)=1$. The authors could include an ablation study or sensitivity analysis to show how the change of these two hyperparameters affect models' performance?

Typo:
line 222: We remain neutral toward the harmful but unhelpful responses like (c) -> (d)
line 223: ... and we hate the harmful yet exhaustive (helpful) responses like (d) -> (c)

---

> ### Author Response · Authors · 2024-11-25
> **Response**
>
> We thank the reviewer for their valuable comments and constructive suggestions.  We are pleased that you found our work novel, and that you recognized the theoretical soundness of our label function. Additionally, we appreciate your acknowledgment of the extensive experiments conducted to validate the soundness and effectiveness of BFPO.  Below, we address each point in detail:
>
>
>
>
> **1. Notations**
>
> We appreciate the reviewer’s detailed feedback regarding the notations. We have revised the manuscript accordingly with improved notation and clearer descriptions. These changes have been marked in blue in the updated version. Specifically, we now use $r(x,y)$ for all reward model, and $g(y|x), g_I(y|x)$ for the functio of reward or labels. The definitions of $\succ$ operation are added at the beginning of section 2.
>
>
> **2. Ablation study or sensitivity analysis**
>
> We thank the reviewer for highlighting the importance of an ablation study. We agree that this is a critical component that can provide deeper insights and enhance the applicability of our method. We will follow section 4.2 to include an ablation study of various pairs of hyperparameters.
>
>
> As mentioned on line 234 of the manuscript, both $B_1$ and $B_3$ must be positive. For this ablation study, we explore the following values for  $B_3= 2， 1，1/2, 1/4$. Given the constraint $B_3(B_1 - 1) = 1$, the corresponding values of $B_1$ are determined for each $B_3$. Additionally, $B_2$ is adjusted to balance the cases described in Figure 3 (Case 2 and Case 3).
>
>
> When $B_3$ is smaller, the label differences for cases 1,2 and 3,4 in Figure 3 become less pronounced. For example, in Cases 1 and 2, the pairs (helpful-safe, non-helpful-unsafe) and (helpful-safe, non-helpful-safe) have smaller differences in their label values. This means there is less distinction in whether the non-helpful response is safe or not.
> As a result, the model shows slightly worse performance in helpfulness but performs better in safety .When $B_3$ is larger, the label differences for the aforementioned two cases become more distinct, and the label value for (helpful-safe, non-helpful-unsafe) becomes significantly higher. This leads the model to prioritize safety more strongly, which results in improved safety performance but a sacrifice in helpfulness.
>
>
> To conclude, larger $B_3$ values emphasize safety at the expense of helpfulness, while proper values allow for more balanced performance across both objectives.
> |  B_3 | B_1 |  B_2  | Values for four cases in Figure 3  | Helpfulness | Harmlessness |       |       |
> |:----:|:---:|:-----:|:----------------------------------:|:-----------:|:------------:|:-----:|:-----:|
> |      |     |       |                                    |    Alpaca   |     Disc     |  Gen  |  Savg  |
> |   2  | 1.5 | -0.25 |         2.5,0.5,-0.5,-0.25         |     9.00    |     58.67    | 95.47 | 77.07 |
> |   1  |  2  |  -0.5 |          1.5,0.5,-0.5,-1.5         |    11.36    |     60.28    | 95.12 | 77.70 |
> |  0.5 |  3  |   -1  |            1,0.5,-0.5,-1           |     13.33   |     59.09    | 95.24 | 77.16 |
> | 0.25 |  5  |   -2  |      0.75， 0.5，-0.5， -0.75      |    13.15    |     59.63    | 94.25 | 76.94 |

---

> > ### Comment · Reviewer_Tsy7 · 2024-11-27
> >
> > Thank you for the reply. Now the notations are clear for me.

---

### Official Review · Reviewer_e56e · 2024-10-29

**Soundness:** 3
**Presentation:** 2
**Contribution:** 3
**Rating:** 8
**Confidence:** 3

**Summary:**

This paper presents a novel supervised optimization approach, BFPO, which balances the safety and helpfulness of a large language model by introducing a labeling function. The effectiveness of the proposed method is demonstrated by the evaluation of  constructed benchmark datasets.

**Strengths:**

1. The proposed Bi-Factorial Preference Optimization (BFPO) method presents a novel approach to balancing safety and helpfulness in the alignment of LLMs. The authors offer a fresh perspective on addressing these often conflicting objectives by integrating direct optimization with multi-objective reinforcement learning principles.
2. BFPO performs well on constructed benchmark datasets and can alleviate the need for manual labeling in LLM fine-tuning.

**Weaknesses:**

1. Limited Real-World Applicability: The experiments are conducted on a synthetic dataset, which may not reflect real-world complexities. The authors should include experiments on more diverse datasets or practical scenarios to demonstrate the robustness and applicability of BFPO in real-world situations.
2. The paper presentation needs to be improved.

**Questions:**

1. Could you elaborate on how your method performs in more complex, real-world scenarios beyond the synthetic dataset used in your experiments? What steps would you take to validate the effectiveness of BFPO in such cases?
2. Could you provide more detailed information on the safety and helpfulness labels used in your dataset? How do you ensure the consistency and reliability of these labels?
3. In Section 3, "Equation (7)"  there is no mention of "g(x,y)".
4. While BFPO eliminates human prompting and labeling to some extent, the ability to capture subtle differences in complex human preferences remains questionable.
5. In Table 1 of the experimental results, why does Alpaca perform better for DPO than BFPO?

---

> ### Author Response · Authors · 2024-11-25
> **Response 1/2**
>
> We thank the reviewer for their valuable comments and questions. Below, we address each point in detail:
>
> **1. Limited Real-World Applicability**
>
> We hope to clarify that our experiments span both synthetic and real-world datasets:
> - Synthetic Dataset (Section 3.4): These experiments validate the effectiveness of our method in controlled scenarios,
> enabling us to isolate and analyze specific behaviors of the proposed approach, which is identical to those published in tier one venues [IPO, GPO].
> - Real-World Datasets (Sections 4.2 and 4.3): In Section 4.2, we demonstrate how to use real-world preference dataset to improve safety and helpfulness of  pre-trained 7B models. In Section 4.3, we explore to improve the safety of a pre-aligned 7B model. Here, we detail the process of leveraging aligned models, public reward models, and public prompt datasets to construct preference data to enhance both safety and helpfulness(lines 459–463).
>
>
> we disagree with the reviewer and we use real world dataset  (PKU-RLHF, UltraFeedback, UltraChat, line 396-402) for training and (CrowS-Pairs, BBQ, WinoGrande, ETHICS, Moral Permissibility, Simple Ethics Questions, ToxicGen, HHH Alignment, AdvBench, Real Toxicity Prompts, ALERT, Alpaca, line 367-377) for evaluation. The use of training and evalutaion dataset is a common practice as in [PKU-RLHF, Llama2, DPO].
>
>
> We kindly ask the reviewer to clarify any specific datasets or practical scenarios they believe could strengthen our evaluation, and we would be glad to explore them.
>
>
> **2. Presentation, Eq 7**
>
> We appreciate the reviewer’s feedback on improving the presentation. We have refined the manuscript for clarity and precision, with revisions highlighted in blue in the updated version. The revisions include 1) improved language in section 1; 2) revised notation in section 2,3, and 3) Improved experiment description in section 4.
>
>
>
>
> **3. detailed information on the safety and helpfulness labels used in your dataset**
>
> We appreciate the reviewer’s question regarding the safety and helpfulness labels in our dataset. All the dataset information can be found in line 396–402. Here we restate them:
>
>
> Helpfulness Dataset: As mentioned in the line 398, we follow [Zephyr] and use [UltraFeedback] as our helpfulness dataset. Each data item in UltraChat consists of a prompt spanning across various tasks, two responses generated by GPT, and a preference label indicating which response is more helpful. We consider all data in this dataset to be safe as the helpfulness dataset has been carefully cured.
>
>
> Safety Dataset: We use the [PKU-SafeRLHF] as safety dataset. Each data item contains a manually designed harmful prompt, two responses generated by their trained Alpaca model, and labels for both helpfulness preference (between the two responses) and safety for each response.
>
>
> Consistency and Reliability of Labels: The data-label pairs in these datasets are unique and appear only once, so there are no issues of inconsistency or duplication in the labeling process.
> Could the reviewers elaborate more on the consistency and reliability issue so that we can provide a more comprehensive response?
>
>
>
>
>
>
> **3. The ability to capture subtle differences in complex human preferences**
>
> We wish to clarify that the primary objective of our work is to balance safety and helpfulness when aligning large language models (LLMs) with human preferences,  irrespectively, if the given data was collected in a manner that accurately captures the complexity of human preferences. The process of constructing a dataset that captures these differences is a complicated task that is well beyond the scope of this paper. Here we aim to, and similarly to a line of prior art, given a dataset assumed to have been collected in a way that captures this complexity of human preferences, can we get the models to train on this dataset while balancing safety/helpfulness.
>
>
>
>
>
>
> If the reviewer concerns that  the use of a single label function could not capture subtle difference, we would like to note that such label function is not unique to our work; all DPO-based methods rely on a label function for efficiency. While capturing subtle differences in complex human preferences is indeed an important and challenging goal, addressing this issue falls outside the scope of our paper. This remains an open problem in the field of LLM alignment with direct optimization.
>
>
> If the reviewer’s concern relates to any specific aspect of how our method captures human preferences within the scope of safety and helpfulness, we kindly request for further clarification so that we can provide a more detailed response.

---

> ### Author Response · Authors · 2024-11-25
> **Response 2/2**
>
> **4. Why does Alpaca perform better for DPO than BFPO?**
>
> As stated in lines 419–420 of the manuscript, training with the DPO objective tends to be more biased toward helpfulness, which is reflected in the higher Alpaca score. However, this bias comes at the cost of safety, where DPO achieves only an average safety score of 49.03—approximately 30 percentage points lower than BFPO.
>
>
> In contrast, BFPO achieves a more balanced trade-off between safety and helpfulness, aligning better with the multi-objective alignment goals of our method.
>
>
>
>
> [IPO] Azar, Mohammad Gheshlaghi, et al. "A general theoretical paradigm to understand learning from human preferences." International Conference on Artificial Intelligence and Statistics. PMLR, 2024.\
> [GPO] Tang, Yunhao, et al. "Generalized Preference Optimization: A Unified Approach to Offline Alignment." Forty-first International Conference on Machine Learning.\
> [Llama2] Touvron, Hugo, et al. "Llama 2: Open foundation and fine-tuned chat models." arXiv preprint arXiv:2307.09288 (2023).
> [DPO] Rafailov, Rafael, et al. "Direct preference optimization: Your language model is secretly a reward model." Advances in Neural Information Processing Systems 36 (2024).
> [PKU-RLHF] Dai, Josef, et al. "Safe RLHF: Safe Reinforcement Learning from Human Feedback." The Twelfth International Conference on Learning Representations.
> [Zephyr] Tunstall, Lewis, et al. "Zephyr: Direct distillation of lm alignment." arXiv preprint arXiv:2310.16944 (2023).
> [UltraFeedback] Cui, Ganqu, et al. "Ultrafeedback: Boosting language models with high-quality feedback." (2023).

---

### Official Review · Reviewer_QMu6 · 2024-10-30

**Soundness:** 3
**Presentation:** 3
**Contribution:** 3
**Rating:** 8
**Confidence:** 4

**Summary:**

This paper presents a supervised learning framework called Bi-Factorial Preference Optimization (BFPO) for enhancing both the safety and helpfulness of large language models (LLMs). By re-parameterizing the multi-objective reinforcement learning from human feedback (RLHF) approach, BFPO integrates the objectives of safety and helpfulness into a single supervised learning objective. In addition, this paper establishes a benchmark based on existing datasets. The experiments demonstrate a superior balance over safety and helpfulness.

**Strengths:**

1. This paper proposes an interesting method to fine tune LLM directly on the preference datasets to balance helpfulness and safety. In addition, the justification and theorems provide a solid foundation.

2. Experimental results show superior performance. In the experiment of table 1, results demonstrate better balance between safety and helpfulness.

**Weaknesses:**

1. The ablation experiments are incomplete. (1) The significant hyperparameter \alpha in the learning objective is only tested in two scenarios: '\alpha=0 only' and '\alpha=0 & no buffer'. However, the setup of \alpha's values in section [3.4 ILLUSTRATIVE EXAMPLES](row 350, \alpha=0.5) and section [4.2 ALIGNMENT WITH BFPO OBJECTIVE](row 394, \alpha=-0.5) differ entirely. This inconsistency prevents a clear understanding of \alpha's true role. (2) The authors do not explore the impact of different values of the hyperparameter \tau, which is also present in BFPO loss function, on the final results.

**Questions:**

Since two hyperparameters, \tau and \alpha, are included in the loss BFPO, could you conduct experiments with a wider range of values for them to test their impact on the final results?

---

> ### Author Response · Authors · 2024-11-24
> **Response**
>
> We thank the reviewer for their valuable comments and suggestions. We are glad that you found our method interesting, and that you appreciated the solid foundation provided by our justifications and theorems. We are also pleased that you recognized the superior performance demonstrated in our experiments.
>
>
> In terms of the hyper-parameters, we did not intentionally tune them to achieve the best performance. The hyperparameters were chosen based on theoretical analysis (Prop 3.3, Figure 3), and we hope BFPO to be robust to these settings.
> Following the reviewer's suggestion, we perform ablation studies on the hyperparameters $\alpha$ and $\tau$ here.
>
>
> **The hyperparameter $\alpha$**
>
>
> The hyperparameter $\alpha$ controls the label values (represent the difference of the preference of a pair of response) of the four cases in Figure 3.
> To ensure the desired behavior—that helpful-safe responses are preferred over helpless-safe ones (Case 2 in Figure 3 yields a positive value) and that helpful-unsafe responses are not preferred over helpless-unsafe ones (Case 3 in Figure 3  yields a negative value)—we constrain $\alpha \in (0, 1)$.
>
>
> When $\alpha = 0.5$, the label values for the four cases are $1,0.5,-0.5,-1$, where the absolute label values are symmetric for positive and negative pairs. As $\alpha$ increases, the absolute label values in case 1,2 in figure 3 decreases, and the absolute label values case 3,4 in figure 3 increases. In other words, the positive pairs will have smaller differences and negative pairs will have larger differences.
>
>
> In the ablation study, we follow the experiment of section 4.2 with $\alpha$ values of $0.1$, $0.3$, $0.5$, $0.7$, and $0.9$ to explore its effects systematically. (line 393 is a typo, and we've corrected it in the revised manuscript)
>
> |       | Helpfulness | Harmlessness |       |       |
> |:-----:|:-----------:|:------------:|:-----:|:-----:|
> | alpha |    Alpaca   |     Disc     |  Gen  |  Savg |
> |  0.1  |    13.61    |     59.81    | 87.39 | 73.60 |
> |  0.3  |    14.06    |     60.31    | 91.73 | 76.02 |
> |  0.5  |     13.33   |     59.09    | 95.24 | 77.16 |
> |  0.7  |     9.01    |     57.34    | 96.28 | 76.81 |
> |  0.9  |     7.21    |     56.44    | 96.66 | 76.55 |
>
> The results show that higher $\alpha$ values reduce distinctions between positive pairs, particularly helpful-safe vs. non-helpful-safe, leading to a lower helpfulness score. However, it increases distinctions between negative pairs, especially helpful-unsafe vs. non-helpful-safe, resulting in improved harmlessness, particularly in generative tasks.
>
>
> **The hyperparameter $\tau$**
>
> The hyperparameter $\tau$ is the coefficient of the KL term in Equation 7, which prevents the policy from deviating from the reference policy (line 120). In practice, it is important to note that $\tau$ is more related to the training and convergence(Figure 4) rather than being a core component of our proposed algorithm to balance the safety and helpfulness.
>
>
> In our experiments, we follow the settings from [Zephyr], where $\tau = 0.01$ is used. This value is applied consistently across all baselines to ensure a fair comparison.
>
>
> For the ablation study, we adopt $\tau$ values inspired by [GPO], specifically $\tau = 0.01, 0.1, 0.5, 1.0$. The results indicate that performance can vary significantly with different $\tau$ values. With different $\tau$, other training hyper-parameters, like the learning rate, training iterations also need to be carefully chosen.
>  |      | Helpfulness | Harmlessness |       |       |
> |:----:|:-----------:|:------------:|:-----:|:-----:|
> |  tau |    Alpaca   |     Disc     |  Gen  |  Savg |
> | 0.01 |     13.33   |     59.09    | 95.24 | 77.16 |
> |  0.1 |     6.4     |     55.44    | 81.45 | 68.44 |
> |  0.5 |     6.53    |     54.01    | 78.14 | 66.07 |
> |  1.0 |     6.74    |     54.10    | 77.52 | 65.81 |
>
>
> [Zephyr] Tunstall, Lewis, et al. "Zephyr: Direct distillation of lm alignment." arXiv preprint arXiv:2310.16944 (2023). \
> [GPO] Tang, Yunhao, et al. "Generalized Preference Optimization: A Unified Approach to Offline Alignment." Forty-first International Conference on Machine Learning.

---

> > ### Comment · Reviewer_QMu6 · 2024-11-25
> >
> > Thank you for your reply. You've answered all my questions.

---

### Official Review · Reviewer_NFRm · 2024-11-08

**Soundness:** 3
**Presentation:** 2
**Contribution:** 2
**Rating:** 6
**Confidence:** 3

**Summary:**

This paper proposes Bi-Factorial Preference Optimization (BFPO) to address the limitations in balancing the helpfulness and safety. Specifically, this paper converts the multi objective RLHF into a modified direct preference alignment, while considering the factor of safety. Through experiments on several alignment datasets including helpfulness and safety, this paper demonstrates the effectiveness of the proposed method.

**Strengths:**

1. This paper converts the multi-objective RLHF objective into a direct preference alignment paradigm that balances the safety and helpfulness.
2. Derived from existing findings, this paper used the global reward and safety label functions to effectively fine-tune the model towards safety and helpfulness via a supervised learning form. Though the notations part of the method is not clear enough, the theoretical part is mostly correct.
3. The experiments include various datasets to demonstrate the effectiveness of the proposed method in improving the model's safety and helpfulness.

**Weaknesses:**

1. This paper overclaims that it does not need human annotations. However, the preference datasets used for BFPO still need the annotations. You can't claim it does not need it only because these datasets are publicly available. Otherwise, nearly all methods can claim they don't need any annotations since there likely existed a prepared dataset before. I hope the authors can revise this part.
2. My major concern is about the writing and annotations of the method section. There are many annotations are not well explained and even confusing. For example, Eq. 5 should have more details instead of citing two papers. L347 is also confusing, $\theta$ is the param of the model, what do you mean be softmax it? How do you define the action space? Also, the term of buffered training is confusing, I don't see the connection between it with buffering in RL. It seems like it just sampled two batches from safety and helpfulness datasets.
3. Question: Why don't the authors adopt LoRA or other existing PEFT methods? Freezing the selected layers of LLM for fine-tuning may be not fair for baseline methods, and often need more validation in the experiments. I hope the authors can validate BFPO with LoRA.
4. Question: what's the difference between your label function and the reward model (including helpfulness and safety) used in Llama? Llama uses piecewise function, and should have similar impacts on training the LLM?

**Questions:**

See the weakness.

---

> ### Author Response · Authors · 2024-11-25
> **Response 1/2**
>
> We thank the reviewer for the valuable comments and suggestions. We are glad that you appreciated our work on converting the multi-objective RLHF objective into a direct preference alignment paradigm, balancing safety and helpfulness. We are also pleased that you found our approach effective, as well as recognizing the strength of our experiments across diverse datasets. We will address the concerns and questions in the following.
>
>
> **1. Overclaiming about the need for human annotations**
> We appreciate the reviewer for highlighting this point. After revisiting our manuscript, we acknowledge that certain statements (e.g., line 083) could be misleading. Our intention throughout the paper was to emphasize that our method achieves comparable performance to approaches requiring extensive prompting, sampling, and human annotations(often known as red teaming), which contributes significantly to the high cost of safety alignment (abstract, lines 051, 092, 505-510). We have revised the language in the manuscript in blue to reflect this intention clearly.
>
>
> Additionally, the reviewer may have overlooked the significance of  the family of direct preference optimization to use publicly available dataset, due to its similar learning process to the supervised learning in other tasks.In preference-based alignment, leveraging an existing dataset is inherently challenging because there is no ground truth to the model outputs, and feedback tied to the model is relied upon to improve its outputs.For example, in RLHF and similar methods, achieving a specific preference X for a model M (e.g., helpfulness) usually requires sampling responses from M according to prompts, having human experts or a reliable scoring model provide feedback based on criteria related to X, and then applying optimization algorithms to refine M. When dealing with multi-objective goals, such as helpfulness and harmlessness, it becomes increasingly difficult to define appropriate prompts, scoring criteria, and feedback mechanisms. In contrast, other tasks like segmentation or summarization often have gold-standard criteria, allowing models to be trained directly against those criteria.
>
>
> This challenge has motivated methods like DPO, IPO, and our approach, to enable the (re)use of public available dataset generated by other models to refine the model being trained.
>
>
> Moreover, even with supervised training, achieving a new preference Y when a dataset for preference X already exists often requires the costly collection of new labels for combinations of X and Y jointly, which incurs significant annotation overhead. In contrast, our method avoids this expense by designing a ranking preference function that merges the properties of two existing preference datasets (X and Y), independently created, without requiring a new annotation process. As a result, our approach allows the creation of a joint preference training framework while leveraging pre-existing datasets focusing on different preference, thereby eliminating the need for new costly annotations and significantly reducing the overhead involved in combining preferences.
>
>
> We hope this explanation, along with revised text (marked in blue in the updated manuscript), addresses the reviewer’s concern.

---

> ### Author Response · Authors · 2024-11-25
> **Response 2/2**
>
> **2. Writing and annotations of the method section**
> - EQ. 5: The derivation of Eq. 5 is detailed in Theorem 3.1 and Appendix B.2. Due to space limitations, we provided a brief summary in the preliminary rather than stating it in full. To improve clarity, we added a connection sentence that links Eq. 5 to its detailed derivation.
> - L347(before update): The target is to learning a parameterized function that maps the actions to their correct probability, such that preferred action will have larger probability to be sampled. The action space is defined as ${y_1, y_2, y_3, y_4}$ (line 340).  $\theta \in \mathbb{R}^4$ represents the values of the action space and the softmax project these values into a probability distribution. For instance, if $\theta$ is initialized with $ [1, 2, 3, 4]$, the policy currently would sample action $y_1$ with probability $ \pi_{\theta} (y_1) = \text{Softmax} (\theta) [1]   = \frac{e^1}{e^1 + e^2 + e^3 + e^4}$. Then the target is to optimize $\theta$ such that $\pi_\theta(y_1) > \pi_\theta(y_3)> \pi_\theta(y_4) > \pi_\theta(y_3)$
>
>
>
>
> We acknowledge that using $\theta$ may cause confusion; however, we aimed to maintain consistency with the original paper [IPO], as this is an identical experiments to theirs where we use it, similarly to them, to verify the loss function on a toyish problem.
> - Buffered training: As mentioned in cited paper (Chaudhry et al. (2019), line 323), "buffered training" is a technique commonly used in continual learning to mitigate catastrophic forgetting. However, we understand that the term may be ambiguous due to its usage in RL. We will rephrase this term to avoid confusion.
>
>
> We are grateful for the reviewer’s detailed feedback on our writing and annotations. All suggested modifications have been implemented and are marked in blue in the revised manuscript. We welcome further feedback.
>
> **3. Why don't the authors adopt LoRA or other existing PEFT methods?**
> This is a great point! While fine-tuning strategies are not the primary focus of our paper, our method is compatible with both full fine-tuning and parameter-efficient fine-tuning (PEFT) methods like LoRA and selective fine-tuning, which involves choosing specific parameters to freeze or update.
>
>
> To ensure a fair comparison, we applied the same fine-tuning strategy across all baselines. We recognize that this detail is better suited for the implementation section rather than the algorithm section, and we have updated the manuscript accordingly.
>
>
> Our decision to fine-tune specific parameters rather than using LoRA was guided by the following considerations:
> 1. Efficiency: While LoRA avoids modifying pre-trained weights, it requires additional parameters, leading to longer back-propagation times in our experiments.
> 2. Convergence Speed: Directly fine-tuning specific parameters rather than involving extra zero-initialized parameters achieves faster convergence, particularly in low-resource settings, as discussed in [A].
> 3. Effectiveness: Modifying only MLP layers is enough to yield targeted behavioral[B,C,D].
>
>
> Following the reviewer's great suggestion, we follow the setting of section 4.2 and conduct additional experiments to compare our method and the best performed baseline, IPO, with LoRA fine-tuning. During the training, we apply the same training hyper-parameters to both algorithms, like learning rate, training epochs, beta, and so on.
> The results are below and will be included in the revised manuscript.
>
>
> Results show that BFPO consistently outperformed the baselines when training with LoRA. However, we observed that LoRA training required additional hyperparameter tuning, which posed challenges due to the limited time. Consequently, both methods achieved lower overall performance and worse balance compared to selective fine-tuning.
>
> |      |     LoRA    |              |       |       |
> |:----:|:-----------:|:------------:|:-----:|:-----:|
> |      | Helpfulness | Harmlessness |       |       |
> |      |    Alpaca   |     Disc     |  Gen  |  Savg |
> |  IPO |     6.14    |     58.05    | 93.97 |  76.1 |
> | BFPO |     7.77    |     64.36    | 94.73 | 79.54 |
>
>
> [A]Empirical Analysis of the Strengths and Weaknesses of PEFT Techniques for LLMs, ICLR-Workshop 2023. \
> [B]Overcoming Generic Knowledge Loss with Selective Parameter Update, CVPR 2024. \
> [C]Locating and Editing Factual Associations in GPT, NeurIPS 2022.  \
> [D]Mass-Editing Memory in a Transformer, ICLR 2023. \
> [IPO]A general theoretical paradigm to understand learning from human preferences. International Conference on Artificial Intelligence and Statistics. PMLR, 2024.

---

> ### Author Response · Authors · 2024-11-26
> **Looking Forward to Your Reply**
>
> Dear Reviewer  NFRm,
>
> Thank you for dedicating your time and effort to reviewing our submission. We posted responses to the concerns raised in your initial reviews and are eagerly await your thoughts. Please let us know if you require any additional information or clarification, and we are happy for further discussions!
>
> Thank you very much!

---

> ### Author Response · Authors · 2024-11-29
> **Looking forward to discussion**
>
> Dear reviewer,
>
> We are looking forward to your response and further discussions. We are happy to resolve any of your concerns and we believe collaboration between dedicated authors and reviewers is essential to producing high-quality research!

---

### Author Response · Authors · 2024-11-25
**General Response**

We sincerely thank the reviewers for their recognition of our efforts in improving the safety of large language models (LLMs). We are encouraged by the positive reception of our motivation to reduce the cost of safety alignment, which has been described as offering a "fresh perspective" (e56e). On seeing the potential of direct preference optimization, we were driven by the belief that a deeper theoretical equivalence for safety alignment lay hidden within, and finally uncover the reparameterization of reinforcement learning-based multi-objective preference optimization into a supervised reward learning framework. We are truly grateful that the reviewers appreciated this effort, describing our theorems as providing a "correct"(NFRm) and "solid foundation"(QMu6). As noted by the reviewers, this theoretical framework translates into "superior performance" (QMu6) demonstrated in our "extensive experiments" (Tsy7). This recognition encourages us to continue pursuing theoretically grounded, high-performance research to improve the safety of foundational models.




In response to the reviewers’ suggestions, we have revised the manuscript with changes marked in blue. These revisions include:
- Enhancing the precision of language to better describe our contribution in efficiency.
- Clarifying the notations and definitions in the formulation.
- Avoiding ambiguous terms and providing more detailed explanations in the experiment. Ensure consistent number format in the results table.




Additionally, as requested, we have conducted ablation studies to provide further insights into our method, indcluding:
- The  impact of hyperparameters $\alpha$ and $\tau$ in the loss function
- The  impact of hyperparameters $B_1$ and $B_3$ in the labeling function
- The impact of different PEFT strategies on the performance of BFPO


Once again, we sincerely thank the reviewers for their thoughtful feedback and careful reading of our work. We welcome further suggestions and remain committed to addressing any future concerns.

---

### Meta-Review · Area_Chair_j5Q1 · 2024-12-10

**Metareview:**

This paper proposes a novel loss function to balance helpfulness and safety in large language model alignment by transforming multi-objective RLHF into a direct preference-based loss. Experiments on various alignment datasets, including those focused on helpfulness and safety, demonstrate the effectiveness of the proposed approach.

The reviewers acknowledge the significance of the proposed loss function and the theoretical analysis supporting its design. The proposed BFPO method demonstrates strong empirical performance on constructed benchmark datasets for safety and helpfulness evaluation. While initial concerns were raised about the clarity of notations in the loss derivation and theoretical analysis, these issues were effectively addressed by the authors during the rebuttal period.

Therefore, we recommend acceptance.

**Additional Comments On Reviewer Discussion:**

The reviewers requested clarification on the notation and derivation of the theorem, which the authors adequately addressed during the rebuttal.

---

### Decision · Program_Chairs · 2025-01-22

Accept (Spotlight)